# OFFLINE RL WITH RESOURCE CONSTRAINED ONLINE DEPLOYMENT

## ABSTRACT

Offline reinforcement learning is used to train policies in scenarios where real-time access to the environment is expensive or impossible. As a natural consequence of these harsh conditions, an agent may lack the resources to fully observe the online environment before taking an action. We dub this situation the *resource-constrained* setting. This leads to situations where the offline dataset (available for training) can contain fully processed features (using powerful language models, image models, complex sensors, etc.) which are not available when actions are actually taken online. This disconnect leads to an interesting and unexplored problem in offline RL: **Is it possible to use a richly processed offline dataset to train a policy which has access to fewer features in the online environment?** In this work, we introduce and formalize this novel resource-constrained problem setting. We highlight the performance gap between policies trained using the full offline dataset and policies trained using limited features. We advocate the use of transfer learning to address this performance gap by first training a teacher agent using the offline dataset where features are fully available, and then transfering this knowledge to a student agent that only uses the resource-constrained features. We evaluate the proposed approach on three diverse set of tasks: MuJoCo (continuous control), Atari 2600 (discrete control) and a real life Ads dataset. Our analysis shows the proposed approach improves over the considered baselines and unlocks interesting insights. To better capture the challenge of this setting, we also propose a data collection procedure: Resource Constrained-Datasets for RL (RC-D4RL).

## 1 INTRODUCTION

There have been many recent successes in the field of Reinforcement Learning (Mnih et al., 2013; Lillicrap et al., 2015; Mnih et al., 2016; Silver et al., 2016; Henderson et al., 2018). In the online RL setting, an agent takes actions, observes the outcome from the environment, and updates its policy based on the outcome. This repeated access to the environment is not feasible in practical applications; it may be unsafe to interact with the actual environment, and a high-fidelity simulator may be costly to build. Instead, *offline RL*, consumes fixed training data which consist of recorded interactions between one (or more) agent(s) and the environment to train a policy (Levine et al., 2020). An agent with the trained policy is then deployed in the environment without further evaluation or modification. Notice that in offline RL, the deployed agent must consume data in the same format (for example,having the same features) as in the training data. This is a crippling restriction in many large-scale applications, where, due to some combination of resource/system constraints, all of the features used for training cannot be observed (or misspecified) by the agent during online operation. In this work, we lay the foundations for studying this *Resource-Constrained* setting for offline RL. We then provide an algorithm that improves performance by transferring information from the full-featured offline training set to the deployed agent's policy acting on limited features. We first illustrate a few practical cases where resource-constrained settings emerge.

**System Latency** A deployed agent is often constrained by how much time it has to process the state of the environment and make a decision. For example, in a customer-facing web application, the customer will start to lose interest within a fraction of a second. Given this constraint, the agent may not be able to fully process more than a few measurements from the customer before making a decision. This is in contrast to the process of recording the training data for offline RL, where one

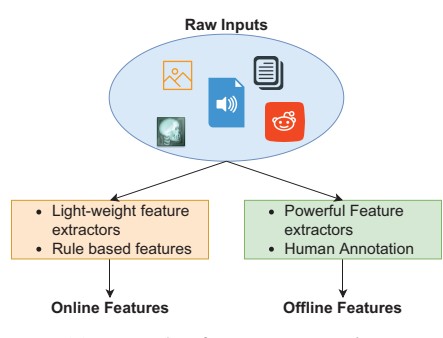

Online Features          Offline Features

(a) Example of system constraints

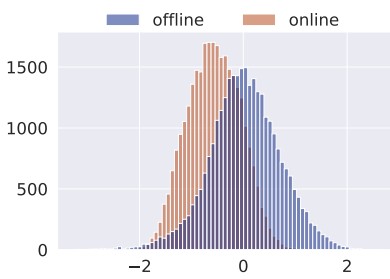

(b) Histogram of rewards in online data collected by agents trained with online versus offline features

Figure 1: Illustrations of the resource-constrained setting

may take sufficient time to generate an abundance of features by post-processing high-dimensional measurements.

**Power Constraints** Consider a situation where an RL agent is being used in deep space probes or nano-satellites ((Deshmukh et al., 2018)). In this case an RL agent is trained on Earth with rich features and a large amount of sensory information. But when the agent is deployed and being used on these probes, the number of sensors is limited by power and space constraints. Similarly, consider a robot deployed in a real world environment. The limited compute power of the robot prevents it from using powerful feature extractors while making a decision. However, such powerful feature extractors can be used during the offline training of the robot (Fig 1a).

In the resource-constrained setting, one can simply ignore the offline features and only train the offline agent with the online features that are available during deployment. This strategy has the drawback of not utilizing all of the information available during training and can lead to a sub-optimal policy. To confirm this, we performed the following simple experiment. We consider an offline RL dataset for the OpenAI gym MuJoCo HalfCheetah-v2 environment and simulate the resource-constrained setting by removing a fixed set of randomly selected features during deployment (see Sections 5.1.1, C.1 for more details). We train an offline RL algorithm, TD3+BC (Fujimoto & Gu, 2021) using only the online features and collect online data in the environment using the trained policy. We repeat this assuming all features available during deployment, train a TD3+BC agent using the same offline dataset with all features, and collect online data in the environment. We plot the histogram of rewards in the two datasets in Fig 1b. We observe that the agent trained only with online features obtains much smaller reward than the agent trained with offline features.

Traditionally, scenarios where the observability of the state of the system is limited are studied under the Partially Observable Markov Decision Process (POMDP) setting by assuming a belief over the observations (Åström, 1965). In contrast, we have an offline dataset (which records rich but not necessarily full state transitions) along with partially obscured (with respect to the offline dataset) observations online. Our goal is to leverage the offline dataset to reduce the performance gap caused by the introduction of resource constraints. Towards this, we advocate using a teacher-student *transfer* algorithm. Our main contributions are summarized below:

- We identify a key challenge in offline RL: in the resource-constrained setting, datasets with rich features cannot be effectively utilized when only a limited number of features are observable during online operation.

- We propose the transfer approach that trains an agent to efficiently leverage the offline dataset while only observing the limited features during deployment.

- We evaluate our approach on a diverse set of tasks showing the applicability of the transfer algorithm. We also highlight that when the behavior policy used by the data-collecting agent is trained using a limited number of features, the quality of the dataset suffers. We propose a data collection procedure (RC-D4RL) to simulate this effect.

## 2 RESOURCE-CONSTRAINED ONLINE SYSTEMS

In the standard RL framework, we consider a Markov Decision Process (MDP) defined by the tuple $(\mathcal{S}, \mathcal{A}, R, P, \gamma)$ where $\mathcal{S}$ is the state space, $\mathcal{A}$ is the action space, $R : \mathcal{S} \times \mathcal{A} \to \mathbb{R}$ is the reward function, $P : \mathcal{S} \times \mathcal{A} \to \Delta(\mathcal{S})$ is the transition function, $\Delta(\mathcal{S})$ denotes the set of all probability distributions over $\mathcal{S}$, and $\gamma \in (0, 1)$ is the discount factor. We consider the *discounted infinite horizon MDP* in this paper. We consider the continuous control setting and assume that both $\mathcal{S}$ and $\mathcal{A}$ are compact subsets of a real-valued vector space. The transition at time $t$, is given by the tuple $(s_t, a_t, R(s_t, a_t), s_{t+1})$. Each policy $\pi : \mathcal{S} \to \Delta(\mathcal{A})$, has a value function $Q^\pi : \mathcal{S} \times \mathcal{A} \to \mathbb{R}$ that estimates the expected discounted reward for taking action $a$ in state $s$ and uses the policy $\pi$ after that. The goal of the agent is to learn the policy $\pi$ that maximizes the expected discounted reward $\mathbb{E}_\pi[\sum_{t=0}^{\infty} \gamma^t R(s_t, a_t)]$. In *online RL*, this problem is solved by interacting with the environment.

In *offline (or batch) RL* (Lange et al., 2012), instead of having access to the environment, the agent is provided with a finite dataset of trajectories or transitions denoted by $D = \{(s_i, a_i, r_i, s'_i)\}_{i=1}^{N}$. The data is collected by one or many behavior policies that induce a distribution $\mu$ on the space of $\mathcal{S} \times \mathcal{A}$. The goal of the agent is to learn a policy using the finite dataset to maximize the expected discounted reward when deployed in the environment.

In the *resource-constrained* setting, the agent does not have access to the full state space or features during deployment. Instead, the agent can only observe from $\widehat{\mathcal{S}}$ (another bounded subset of the real-valued vector space) that is different from $\mathcal{S}$. It is assumed that the space $\mathcal{S}$ is rich in information as compared to $\widehat{\mathcal{S}}$. For example, $\widehat{\mathcal{S}}$ might have fewer dimensions, or some entries may include extra noise (see Figure 1a). We will use *online/limited features*, to refer to observations from the online space $\widehat{\mathcal{S}}$, *offline/rich features* to refer to observations from the offline space $\mathcal{S}$. We assume that both online features and offline features are available in offline data.

The goal of the agent is to use the offline data and train a policy $\pi : \widehat{\mathcal{S}} \to \Delta(\mathcal{A})$. The agent can use the offline features from $\mathcal{S}$ during training but is constrained to only use the online features from $\widehat{\mathcal{S}}$ while making a decision. A similar paradigm of Learning Under Privileged Information (LUPI) (Vapnik et al., 2015) has been studied under the supervised learning setting, where the privileged information is provided by a knowledgeable teacher.

## 3 RELATED WORK

**Offline RL** There has been an increasing interest in studying offline RL algorithms due to its practical advantages over online RL algorithms (Agarwal et al., 2020; Wu et al., 2021; Chen et al., 2021; Brandfonbrener et al., 2021). Offline RL algorithms typically suffer from overestimation of the value function as well as distribution shift between the offline data and on-policy data. Buckman et al. (2020) and Kumar et al. (2020) advocate a pessimistic approach to value function estimation to avoid over-estimation of rarely observed state-action pairs. To constrain the on-policy data to be closer to offline data, several techniques have been explored, such as restricting the actions inside the expectation in the evaluation step to be close to the actions observed in the dataset (Fujimoto et al., 2019), adding a regularization term during policy evaluation or iteration (Kostrikov et al., 2021), (Wu et al., 2019),(Guo et al., 2020), adding a constraint of the form $\text{MMD}(\mu(\cdot|s), \pi(s))$ (Gretton et al., 2012; Blanchard et al., 2021; Deshmukh et al., 2019) on the policy (Kumar et al., 2019), using behavior cloning (Fujimoto & Gu, 2021), adding an entropy term in the value function estimation (Wu et al., 2019), and model-based approaches that learn a pessimistic MDP Kidambi et al. (2020). A thorough review of these techniques is presented in an excellent tutorial by Levine et al. (2020). To the best of our knowledge, there is no existing work that addresses the resource-constrained offline RL setting where there is a mismatch between the offline features and online features.

**Knowledge Transfer** *Knowledge transfer/distillation* is widely studied in various settings including vision, language, and RL domains (Gou et al., 2021; Wang & Yoon, 2021). In RL, under the *domain transfer* setting (Taylor & Stone, 2009; Liu et al., 2016), the teacher is trained on one domain/task and the student needs to perform on a different domain/task (Konidaris & Barto, 2006; Perkins et al., 1999; Torrey et al., 2005; Gupta et al., 2017). Li et al. (2019) train a model so that features from different domains have similar embeddings, and Kamienny et al. (2020) perturb the feature using a random noise centered at the privileged information. An offline RL algorithm for domain transfer

has been proposed by Cang et al. (2021). *Policy distillation* is studied in the setting where the knowledge from a trained policy (teacher) is imparted to an untrained network (student) (Rusu et al., 2015; Czarnecki et al., 2019). This leads to several advantages such as model compression and the ability to learn from an ensemble of trained policies to improve performance (Zhu et al., 2020).

One distinguishing feature of the resource-constrained setting that differentiates it from other transfer settings is that the teacher has access to the privileged information and student needs to adapt from the data available without interactive learning. In most of the existing approaches, the difference between teacher and student was either the network size (which is also present in our setting due to the difference in input features) or the dynamics (as in the domain transfer case). To the best of our knowledge, we are the first to study policy distillation in the offline RL framework under the resource-constrained setting. Another interesting line of work is called Sim2Real (Lee et al., 2021; Traoré et al., 2019). In these papers, they train a model using a simulator and then transfer the knowledge to real data. However, this work requires an accurate simulator which results in a fairly expert teacher model. However, in the offline RL setting, depending on the data quality, the teacher itself might be weak.

**Partially Observable MDP** POMDP generalizes the MDP framework where the agent does not have access to the full features and only partially observes the state space (Åström, 1965; Kaelbling et al., 1998; Ortner et al., 2012). More recently, Rafailov et al. (2021) studied a model-based offline RL algorithm for image data under the POMDP setup. Our setting resembles this setup, but our agent also has access to the full privileged features in the offline dataset while training. This availability of the offline dataset with privileged information differentiates our setting and enables the student to inherit the knowledge from the rich space while only using the limited features during deployment.

## 4 PROPOSED ALGORITHM

In this section, we advocate the use of transfer algorithms to address the resource constrained setting. In particular, we discuss teacher-student knowledge transfer models.

### 4.1 CONTINUOUS CONTROL

We motivate the proposed algorithm by first providing a general approach to tackle the resource-constrained setting using policy distillation (Rusu et al., 2015; Czarnecki et al., 2019). This is a modeling choice; alternative types of knowledge transfer, such as transfer of the value function's knowledge, are also possible. (Czarnecki et al., 2019).

We first train a teacher network using the full offline dataset. Let us denote the policy output by the teacher network as $\pi_{\phi^{\text{teacher}}} : \mathcal{S} \to \Delta(\mathcal{A})$. The student network is denoted by $\pi_\phi : \widehat{\mathcal{S}} \to \Delta(\mathcal{A})$. The knowledge transfer is performed by using the pre-trained teacher network to compute a regularization term that is added to the objective (policy iteration):

$$\pi_{k+1} \leftarrow \arg\max_\pi \mathbb{E}\left[\widehat{Q}^\pi_{k+1}(\hat{s}, a) - \beta\mathcal{M}\Big(\pi_\phi(\hat{s}), \pi_{\phi^{\text{teacher}}}(s)\Big)\right]. \tag{1}$$

Here, $\mathcal{M}$ can be any divergence metric on the policy distributions. We can compute the terms $\pi_\phi(\hat{s})$ and $\pi_{\phi^{\text{teacher}}}(s)$ simultaneously since we assume that the offline dataset contains the offline features $s$ and the corresponding online features $\hat{s}$. If the teacher and learned policy functions are deterministic, the divergence function can be the squared loss, and if they are stochastic, the divergence can be KL divergence, MMD (Maximum Mean Discrepancy), Wasserstein divergence, etc. This regularization term encourages the learnt policy to behave similarly to the teacher policy. An illustration of the proposed setup is given in Figure 6 in the Appendix.

**Proposed Algorithm** We now propose our transfer algorithm that adopts TD3+BC (Fujimoto & Gu, 2021) to the resource-constrained setting. We add an additional regularization term during the policy iteration step, that tries to keep actions predicted by the trained policy close to the actions taken by the teacher policy. This knowledge can be imparted to the student policy through regularization.

$$\arg\max_\phi \frac{1}{N}\sum_{i=1}^N \left[\underbrace{\lambda Q_{\theta_1}(\hat{s}_i, \pi_\phi(\hat{s}_i)) - \beta_1\Big(\pi_\phi(\hat{s}_i) - a\Big)^2}_{\text{TD3+BC}} - \underbrace{\beta_2\Big(\pi_{\phi^{\text{teacher}}}(s_i) - \pi_\phi(\hat{s}_i)\Big)^2}_{\text{Transfer}}\right] \tag{2}$$

We weigh the two terms with weights $\beta_1, \beta_2$. The transfer term is beneficial in learning because the teacher (trained using offline RL algorithm on full features) often predicts a better action than those available in the dataset (depending on the quality of the dataset) as observed in earlier offline RL works (Kumar et al., 2020). To keep all terms in equation 3 at comparable magnitudes, we add a constraint that $\beta_1 + \beta_2 = 1$. If $\beta_1 = 1$ and $\beta_2 = 0$, the proposed algorithm is equivalent to the TD3+BC algorithm. Algorithm 1 describes the complete steps.

## 4.2 DISCRETE CONTROL

For discrete control tasks, Deep Q learning is a widely followed approach (Mnih et al., 2015) and more recently distributional versions of deep Q learning are considered state of the art (Dabney et al., 2018). For the setting of Offline RL, CQL (Kumar et al., 2020) computes a conservative estimate of the Q function values to minimize overestimation error. The objective minimized by CQL adds an additional conservative loss term to the loss of DQN as follows

$$L(\theta) = L_{DoubleDQN}(\theta) + L_{CQL}(\theta)$$

where

$$L_{DoubleDQN}(\theta) = \mathbb{E}_{s_t,a_t,r_{t+1},s_{t+1}\sim D}[(r_{t+1} + \gamma \underbrace{Q_{\theta'}(s_{t+1}, \mathrm{argmax}_a Q_\theta(s_{t+1}, a))}_{y_{\text{target}}} - Q_\theta(s_t, a_t))^2],$$

$$L_{CQL}(\theta) = \alpha \mathbb{E}_{s_t \sim D}[\log \sum_a \exp Q_\theta(s_t, a) - \mathbb{E}_{a \sim D}[Q_\theta(s_t, a)]]$$

and $\theta$ are the parameters of the Q function and $\theta'$ are the parameters of the target Q function. The original CQL paper uses DQN instead of the DoubleDQN but following Seno & Imai (2021) we consider the Double DQN.

**Proposed Method**: We use the CQL algorithm and propose the transfer algorithm by using the offline teacher's Q function ($Q^{\text{teacher}}$) in computing the $y_{\text{target}}$.

$$y_{\text{target}} = (1 - \beta) \underbrace{Q_{\theta'}(s_{t+1}, \mathrm{argmax}_a Q_\theta(s_{t+1}, a))}_{\text{Argmax uses Student Q}} + \beta \underbrace{Q_{\theta'}(s_{t+1}, \mathrm{argmax}_a Q^{\text{teacher}}(s_{t+1}, a))}_{\text{Argmax uses Teacher Q}}$$

where $\beta$ is a fixed value between [0,1]. This enables the $y_{\text{target}}$ value to be queried at actions that are preferred by the teacher agent at the current state. This can imply updating more frequently the Q values of the state action pairs experienced by the teacher. When $\beta = 0$, this is the same as the CQL algorithm, and when $\beta = 1$, only the $\arg\max Q^{\text{teacher}}(s, \cdot)$ is used to query the student's target Q function.

## 5 EXPERIMENTAL RESULTS

We performed experiments on a variety of domains to study the resource constrained setting. For continuous control tasks, we used the OpenAI gym MuJoCo (Todorov et al., 2012); for discrete control tasks, we used the Atari-2600 environment (Bellemare et al., 2013) and for real world task we consider the task of Auto-bidding and use a proprietary dataset.

## 5.1 MUJOCO

Before we discuss the resource-constrained setup used for the MuJoCo environments in Section 5.1.1, it is important to note that the data-collecting agent plays a vital role in determining the quality of the dataset (coverage of state-action pairs) and subsequently the performance of any offline RL algorithm. In the D4RL suite (Fu et al., 2020), datasets were collected using behavior policies of varying expertise to simulate the variation in data quality. Each of these behavior policies was trained online using the full feature set. Thus, these policies can use all of the information and explore the environment online. The quality of the offline dataset collected by these behavior policies is thus relatively high thereby improving the performance of offline RL algorithms on them. In the resource-constrained setting, however, the behavior policy of the data-collecting agent does not have access to the full features online. The agent may only explore and navigate using the

limited feature set during training. Using this behavior policy to collect data results in a relatively lower quality dataset. To account for this limitation, and to illustrate its effect on our algorithm's performance, we used two different methods to generate offline datasets.

- Offline RL datasets are collected by agents trained with limited features. We refer to these datasets as *RC-D4RL* (Resource-Constrained Datasets for RL). We use the OpenAI gym MuJoCo locomotion environments Hopper-v2, HalfCheetah-v2, and Walker2d-v2.
- Offline RL datasets are collected by agents trained using the full features. We use the D4RL-v0 datasets (Fu et al., 2020) of the MuJoCo locomotion environments.

### 5.1.1 SIMULATION OF RESOURCE-CONSTRAINED SETTING

We simulate the resource-constrained setting by reducing the feature space available during the deployment of the agent. We do this by dropping a fixed set of features from the full features available (this is possible in the system latency as well as the nano-satellite example). For instance, consider the MuJoCo environment Hopper-v2 where the original state space is 11 dimensional. We consider four scenarios where the online observable feature dimensions is reduced to 5, 7, 9 and 10 by randomly picking a subset of the features of the given dimension. For each of these scenarios, we consider 5 random seeds to simulate different features getting dropped in each seed. We summarize this setting for the three environments in Table 1. For the dataset collection using policies trained with limited features, we follow a similar protocol as the D4RL data collection procedure (Fu et al., 2020). We have added all details to Appendix C.1.

| Environment | Original Dim | Resource-Constrained Dim | Number of Seeds |
|---|---|---|---|
| Hopper | 11 | 5, 7, 9, 10 | 5 |
| HalfCheetah, Walker2d | 17 | 9, 11, 13, 15 | 5 |

Table 1: Resource-Constrained Simulation using gym MuJoCo tasks

### 5.1.2 TRAINING AND EVALUATION

We train a teacher agent using the full feature space in the dataset. We use the TD3+BC algorithm trained on the limited feature set as the baseline (unless otherwise mentioned). Then we train the student agent using the trained teacher following Algorithm 1. We train the baseline and student for three random seeds. To illustrate the effect of the transfer, we consider two representative settings, (i) *Transfer (0.5, 0.5)* where $\beta_1 = \beta_2 = 0.5$ which gives equal weight to behavior cloning and transfer, (ii) *Transfer (0.0, 1.0)* where $\beta_1 = 0$, $\beta_2 = 1.0$ where only transfer is performed. We adopted the base hyperparameters from TD3 since hyperparameter tuning in offline RL is a difficult task without the access for the environment during training (discussed in Appendix C). We used the normalized score (Fu et al., 2020) for evaluating the algorithm (more details in Appendix C.2).

From Figures 2 and 3, we can see that the proposed transfer algorithm significantly outperforms the baseline for both the RC-D4RL and D4RL datasets. In Figures 2a and 3a, we show the percentage of experiments (all the difficulties, dimensions, dataset seeds, algorithm seeds) where each of the proposed algorithms performs better than the baseline for each environment. We observe that the proposed algorithm performs similarly for both the datasets with the performance in D4RL being slightly better. In Figures 5d and 3b, we average the scores of all experiments for each environment and algorithm pair and compare the % improvement over the baseline. We also study the loss in performance by not using offline features in Table 2 (more detailed results in Table 5) which shows that the transfer approach is more suitable to the low data quality regime. We perform a thorough analysis of the results and summarize the findings in Appendix C.3, focusing on the effect of difficulties of the datasets and different (limited) feature dimensions.

Additionally, we also train two more baselines: (i) True-BC agent that learns from the teacher policy directly through behavior cloning, (ii) a predictive baseline where an autoencoder is first used to predict the offline features from the observed online feature and uses the predicted features during training and deployment (discussed in detail in Appendix C.4). We evaluated these algorithms on

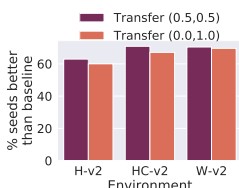 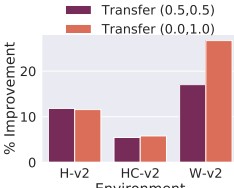

(a) % of succes over the baseline across all experiments.

(b) % improvement over the baseline score across all experiments.

Figure 2: Results on RC-D4RL datasets. [1]

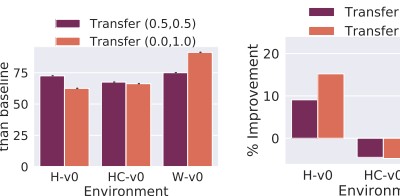

(a) % of succes over the baseline across all experiments.

(b) % improvement over the baseline score across all experiments.

Figure 3: Results on D4RL datasets. [2]

| Difficulty | Baseline (TD3+BC) | Transfer (0.5,0.5) | Transfer (0.0,1.0) |
|---|---|---|---|
| expert | -29.8 % | -32.5 % | -39.6 % |
| medium-replay | -33.2 % | -22.4 % | -14.1 % |

Table 2: We summarize the loss in performance by not using the offline features for the Baseline (TD3+BC), Transfer (0.5,0.5) and (0.0,1.0) as a % change over the Teacher score on RC-D4RL HalfCheetah-v2.

| Difficulty | True-BC | Predictive |
|---|---|---|
| expert | -32.1 % | -18.5% |
| medium-replay | 30.7 % | 31.0% |

Table 3: Average % improvement of the proposed method over the True-BC and Predictive baselines on RC-D4RL HalfCheetah-v2 datasets

RC-D4RL HalfCheetah-v2 medium-replay and expert datasets and present the result summary in Table 3 (more detailed results in Table 6).

## 5.2 ATARI-2600

We evaluate our proposed approach on the Atari 2600 suite of games Bellemare et al. (2013), which is a discrete control problem that is challenging especially due to the high dimensional visual input and delayed credit assignment. Since we are in the offline RL setting, we use the DQN replay dataset from Agarwal et al. (2020) which is a standard benchmark to evaluate offline RL algorithms for discrete control problems. In particular, we use the data for the following games: Pong, Qbert, Breakout.

### 5.2.1 DATASETS

The DQN replay dataset consists of 50M state transitions that are collected during the training of a DQN agent in the online setting. Each transition consists of a tuple (state, action, reward, next_state) where the state and next state are 84x84 images and the action space is discrete. Following earlier works, we consider environments with sticky actions (the agent takes current action with probability 0.25, or otherwise repeats past actions). Additionally, we use stacking of 4 consecutive frames together as the state space which is a common technique used for the Atari suite. Instead of using the full 50M transitions, we use 1M transitions in the dataset that are collected during the end of the DQN online training and hence contain expert level trajectories.

To simulate the online and offline features, we consider the following setup. We assume the actual image observation (4x84x84) as the offline features, and a pixelated version of the images as online features. To simulate this, we resize each (84x84) image to 16x16 and then resize it back to 84x84. This setup reasonably imitates the nano-satellite example where some cheap sensors need to be used online as compared to high resolution sensor data available offline.

We trained the algorithms with a batch size of 32 for 1M batches using the same network architecture as the DQN (Mnih et al., 2015) (which is also used in Agarwal et al. (2020), Kumar et al. (2020)). We refer to it as (N-DQN). We evaluate during training at a frequency of every 20k batches by assuming access to the environment, and consider the score as the average reward achieved in the

---

[1] H-v2: Hopper-v2, HC-v2: HalfCheetah-v2, W-v2: Walker2d-v2

[2] H-v0: Hopper-v0, HC-v0: HalfCheetah-v0, W-v0: Walker2d-v0

last 10 evaluations. The algorithm (from Section 4.2) is implemented using d3rlpy (Seno & Imai, 2021) and use CQL (trained using online features) as the baseline.

In order to simulate the power constrained setting (as in the nano-satellite example), we also consider the setup where the agent to be deployed online uses a different network architecture than the teacher network architecture (N-DQN). This enables the agent to utilize lesser memory and power during deployment. We refer to this setting as the power constrained setting. We call this network architecture as (S-DQN) by changing the size of features in the final layer. Specifically, we vary the feature dimension with values 256, 128, 64 (see Appendix B) for more details.

We observe from both the settings that there is a significant drop in performance of the Baseline and Transfer agents using the online features as compared to the Teacher. The Transfer agent performs marginally better than the Baseline agent on all the three games for the N-DQN setting. For the S-DQN setting, the Transfer agent outperforms the Baseline for Pong on all the three encoder sizes considered. For Qbert, the Transfer agent outperforms the Baseline for one of the encoder sizes. It is important to highlight that there is a significant gap between the performance of the Teacher and the Transfer/Baseline agents for both the settings. This suggests that more tailored transfer learning approaches are required for the Resource Constrained Offline RL problem.

| Game | Size of last layer (#Parameters of network) | Teacher | Transfer | Baseline |
|---|---|---|---|---|
| Pong | 512 (1.6 M) | $8.2 \pm 1.89$ | $-1.02 \pm 1.1$ | $-2.81 \pm 1.39$ |
| Pong | 256 (0.88 M) | - | $-1.08 \pm 0.88$ | $-1.74 \pm 4.83$ |
| Pong | 128 (0.48 M) | - | $-1.83 \pm 4.75$ | $-7.99 \pm 1.52$ |
| Pong | 64 (0.28 M) | - | $-3.53 \pm 5.01$ | $-5.25 \pm 7.95$ |
| Qbert | 512 (1.6 M) | $7890.25 \pm 1896.22$ | $6343.92 \pm 346.45$ | $5833.58 \pm 584.74$ |
| Qbert | 256 (0.88 M) | - | $4762.95 \pm 542.23$ | $4710.00 \pm 73.86$ |
| Qbert | 128 (0.48 M) | - | $2524.80 \pm 717.82$ | $2808.92 \pm 1577.74$ |
| Qbert | 64 (0.28 M) | - | $1336.85 \pm 566.79$ | $1830.00 \pm 908.12$ |
| Breakout | 512 (1.6 M) | $106.63 \pm 8.64$ | $6.07 \pm 0.08$ | $4.65 \pm 0.22$ |

Table 4: Results of the transfer algorithm on three Atari games using N-DQN architecture and the power constrained setting (using S-DQN). We present the mean and std for 3 random seeds

## 5.3 AUTOBIDDING - ADS DATA

Purely simulated settings may not be representative for evaluating an algorithm which is meant for real-world application. Thus, we studied agent performance in the auto-bidding task for online advertising (Bottou et al., 2013). Online advertising is a dynamic, stochastic environment. Advertisers have increasingly delegated decisions to machines in order to achieve increased return on investment. Auto-bidding is one of the critical components in this shift towards AI-driven optimization. Auto-bidding agents determine a unique bid for each opportunity in real-time. This problem is one of distributed, stochastic control in a partially observable, stochastic, non-stationary environment. This type of environment is extremely difficult to develop complex algorithms for, and it demonstrates our setting well in that there are a myriad of computational constraints that limit the type of models that can be considered during online operation.

In this task, agents attempt to maximize the number of ad clicks they collect during a day by bidding on queries (on which ads are shown). The set of queries which receive a bid are set by the advertiser's choice of bidded keywords. Agents are given a fixed daily budget to do this. They are charged according to some black-box auction mechanism only if their ad is clicked. Agents must balance between saving budget for future opportunities, and buying guaranteed ad space now. At each time step, an auto-bidding agent has some information available like time of the query, details of the ad text or bidded keyword, available budget, past spend, model-based estimate of how likely the consumer is to click an ad, etc. This information is used as a state. Based on the state, the agent takes an action (decides the bid) and gets feedback (if the ad was selected and/or clicked, and how much the click costs). If the ad was clicked, the "available budget" feature in the state is updated.

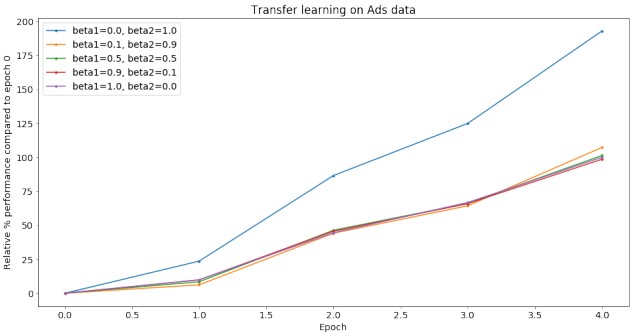

Figure 4: FQE estimated rewards on the Ads data. Only the relative metrics are presented to hide business-sensitive information. 10 different seed values are tried, and the average values are presented. *Transfer (0.0, 1.0)* outperforms the baseline, *Transfer (1.0, 0.0)*, 7 times out of 10.

The budget constraint in this problem means that it cannot be modeled using contextual bandits (as many Ads problems can), because the next state depends on the action taken. If the agent bids $10 given an available budget of $100, it can expect to have at least $90 to spend for the rest of the day. This $90 is part of the state for the next step. If the agent bids $1 in the same situation, it may expect to have at least $99 for the rest of the day. This rest-of-day budget constraint is a key part of the state, as it restricts the trajectory of future states and actions thus, requiring an MDP formulation.

We have pulled query-level Ads data for 10 days and 8,000 advertisers (from a proprietary dataset). The features that are available online and offline include (but not limited to) the time of the query, a model-based estimate of how likely the consumer is to click an ad, remaining budget, and past spend. The features that are only available offline are model-based embeddings of the query and the bidded keywords as they require extensive computational resources. There are 881 features available offline and 111 features available online. The main constraint limiting the types of models and features used in the online setting is computation time, as the user who generated the query will not accept a wait time of more than a fraction of a second before they want to see their search results. We trained on 80% of the advertisers using the proposed TD3+BC transfer algorithm and evaluated on the remaining 20% using the FQE (Le et al., 2019) algorithm.

In Figure 4, FQE estimates are normalized by the value estimate at the beginning of the first epoch (before policies are trained), which is the same value for all settings (since before training all policies are initialized as random with a fixed seed). We report the average normalized metrics across 10 seeds. Normalization is done to mask the identities of advertisers, which are business-sensitive. *Transfer (0.0, 1.0)* tends to perform better than other models including the baseline (*Transfer (1.0, 0.0)*,i.e. without transfer). Furthermore, out of 10 different seeds, *Transfer (0.0, 1.0)* outperforms *Transfer (1.0, 0.0)* 7 times suggesting the applicability of incorporating transfer learning.

## 6 CONCLUSION

In this work, we address an open challenge in Offline RL. In the resource-constrained setting that is motivated by real world applications, the features available during training offline may be different than the limited features available online during deployment. We highlighted a performance gap between offline RL agents trained using only the online features and agents trained using all the offline features. To bridge this gap, we proposed a student-teacher-based policy transfer learning approach. The proposed algorithm improves over the baseline significantly even when the dataset quality is lacking in the resource-constrained setting. The simplicity of the approach (with just one additional hyperparameter) makes it easy to extend it to other offline RL algorithms. It would be interesting to study other transfer learning approaches (e.g., policy transfer with other divergence regularizations for stochastic policies) in the future. Moreover, we observe that the proposed approach benefits especially when the dataset quality is low which is often the case with real world datasets. Despite this, the performance gap with the teacher is still high (in the low quality data regime) and this suggests more tailored approaches are required.

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

## A    ALGORITHM

**Background of TD3+BC** We discuss a simple way to extend an existing offline RL algorithm to apply to the resource-constrained setting. TD3+BC (Fujimoto & Gu, 2021) is a recent offline RL algorithm which shows that by adding an additional behavior cloning term to an existing online RL algorithm TD3 (Fujimoto et al., 2018) (that learns a deterministic policy), it is possible to attain state-of-the-art performance on the D4RL benchmark datasets (Fu et al., 2020). The policy evaluation step in TD3+BC is the same as TD3 where the minimum of two $Q$ functions is used to reduce the over-estimation bias of the value function. In the policy iteration step, the behavior cloning term $(\pi_\phi(s) - a)^2$ regularizes the policy to take actions similar to the actions observed in the dataset. The policy iteration step is given by

$$\pi_{k+1} \leftarrow \arg\max_\pi \mathbb{E}\left[\lambda\widehat{Q}^\pi_{k+1}\big(s, \pi(s)\big) - \big(\pi_\phi(s) - a\big)^2\right]; \qquad \lambda = \frac{\alpha}{\frac{1}{N}\sum_{(s_i, a_i)}|Q(s_i, a_i)|},$$

where $\alpha$ is a hyper-parameter.

---

**Algorithm 1** Policy Transfer with TD3+BC

---

1: **Given:** offline dataset $D$ with full feature observations, the weights $\beta_1, \beta_2$ s.t. $\beta_1 + \beta_2 = 1$
2: **Given:** policy update frequency d; weighted average parameter $\tau$, noise parameter $\bar{\sigma}$
3: Train an offline RL agent using TD3+BC which outputs the teacher model $\pi_{\phi^{\text{teacher}}}$
4: Initialize critic networks $Q_{\theta_1}, Q_{\theta_2}$ and an actor network $\pi_\phi$
5: Initialize target networks $\theta'_1 \leftarrow \theta_1, \theta'_2 \leftarrow \theta_2, \phi' \leftarrow \phi$.
6: **for** $t = 1, \cdots, T$ **do**
7:     Sample mini-batch of N transitions $(s, a, r, s') \in D$
8:     Online features for this transition is given as $(\hat{s}, a, r, \hat{s}')$
9:     $\tilde{a} \leftarrow \pi_{\phi'}(\hat{s}') + \min(\max(\epsilon, -c), c)$ where $\epsilon \sim \mathcal{N}(0, \tilde{\sigma})$
10:    $y \leftarrow r + \gamma\min_{i=1,2} Q_{\theta'_i}(\hat{s}', \tilde{a})$
11:    Update critics $\theta_i \leftarrow \arg\min_{\theta_i} \frac{1}{N}\sum_{i=1}^N \big(y - Q_{\theta_i}(\hat{s}, a)\big)^2$
12:    **if** t mod d == 0 **then**
14:        Update $\phi$ by deterministic policy gradient optimizing the objective

$$\arg\max_\phi \frac{1}{N}\sum_{i=1}^N \left[\lambda Q_{\theta_1}(\hat{s}_i, \pi_\phi(\hat{s}_i)) - \beta_1\Big(\pi_\phi(\hat{s}_i) - a\Big)^2 - \beta_2\Big(\pi_{\phi^{\text{teacher}}}(s_i) - \pi_\phi(\hat{s}_i)\Big)^2\right] \quad (3)$$

15:        Update target networks:
16:        $\theta'_i \leftarrow \tau\theta_i + (1 - \tau)\theta'_i$
17:        $\phi' \leftarrow \tau\phi + (1 - \tau)\phi'$
18:    **end if**
19: **end for**

---

## B    ATARI EXPERIMENT DETAILS

The architecture of N-DQN is given by `[[32,8,4], [64,4,2], [64,3,1], 512]`, where 512 is the size of the final layer. We simulate the power constrained setting by varying the final layer size as 256, 128, 64, thereby reducing the number of parameters of the network and hence the memory and power usage.

The hyperparameters are different from the implementation of CQL (available at https://github.com/aviralkumar2907/CQL). We use QR-DQN Dabney et al. (2018) to compute the DQN loss using 32 quantiles. We use learning rate $6.25 \times 10^{-5}$ and used Adam optimizer with $\epsilon = 1/32 \times 10^{-2}$. The target update interval is set to 8000. The $\alpha$ in CQL loss is set to 1.0. During evaluation, the agent follows an $\epsilon$ greedy approach with $\epsilon = 0.001$.

We selected the value of $\beta$ by choosing the value that resulted in the best performance on Qbert and used it for the other environments. We did this separately for the N-DQN setting and the power constrained setting (for final layer size=256). In particular, we use $\beta = 0.8$ for N-DQN (using 512 in final layer) and we use $\beta = 0.95$ for S-DQN (using 64, 128, 256).



(a) Pong original image   (b) Pong pixelated image   (c) Qbert original image   (d) Qbert pixelated image

Figure 5: Simulation of high resolution and low resolution sensors.

During evaluation, the environment outputs pixelated images that are used by the Baseline and Transfer agents. The pixelated images can be see as shown in Figure 5.

## C  MuJoCo Experiment Details

We used the open source implementation of TD3+BC available at TD3+BC for our experiments. We used the same network architectures, and all other hyperparameters in this implementation. We also normalized the offline RL dataset using the procedure described in Fujimoto & Gu (2021).

Recent work (Fujimoto & Gu, 2021) highlighted the difficulty in comparing algorithmic contributions in Offline RL due to the high sensitivity of deep RL algorithms to implementation changes (Engstrom et al., 2019; Furuta et al., 2021). It is even more crucial in offline RL because hyperparameter tuning by interacting with the environment defies the purpose of offline RL, and tuning without environment access is not well understood (Paine et al., 2020). Therefore, to highlight our algorithmic contributions, we adopt the base hyperparameters of TD3+BC for the baseline and the proposed algorithm.

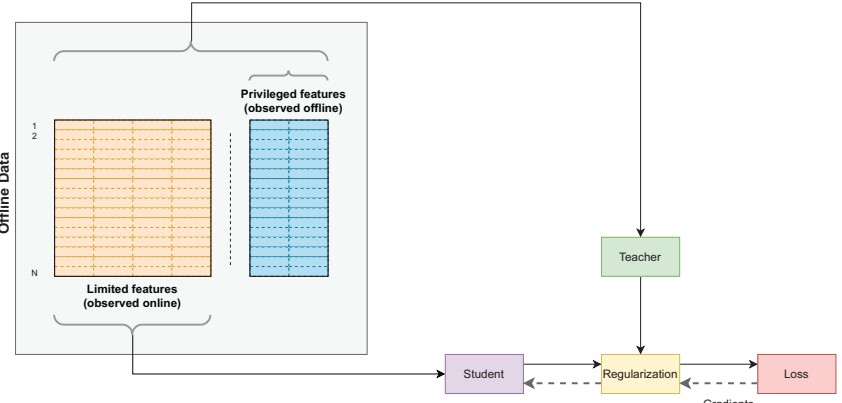

Figure 6: Illustration of the proposed method

### C.1  Dataset collection procedure: RC-D4RL

For the dataset collection using policies trained with limited features, we follow a similar protocol as the D4RL data collection procedure (Fu et al., 2020). In the D4RL benchmark suite, offline datasets with different difficulty levels are collected based on the exploration capacity of the data-collecting policy or behavior policy. The difficulty levels are denoted as medium-replay, medium, medium-expert and expert. For each combination (environment, difficulty, dimension, seed) in Table 1, we train an expert policy by training a TD3 (Fujimoto et al., 2018) algorithm for 1 million timesteps using the limited features, and using the base hyperparameters of the TD3 algorithm. We then train the medium policy by training the TD3 algorithm until the agent reaches roughly half of the reward achieved by the expert policy for this combination. These policies are deployed in the simulator and the interactions are logged to generate the expert and medium datasets. Note that all the features are logged in the datasets. The medium-replay data comprises of the environment interactions recorded

during the training of the medium policy. The medium-expert dataset is the concatenation of the medium and expert policies. Thus, we generate in total 240 [3] offline RL datasets. Note that we did not perform hyperparameter tuning to compute the expert data collection policy and only used the base hyperparameters used in the TD3 paper[4]. We can observe the variation in the quality of the datasets with the dimensionality of online features and the difficulty from Table C.6 which reinforces the need to evaluate the proposed algorithm with the RC-D4RL datasets.

## C.2 EVALUATION

When reporting the performance of the algorithm, we assume online access to the gym MuJoCo simulator for evaluation. The agents are restricted to using the limited features (defined by the combination of the offline dataset used during training) to make a decision. Given a policy to evaluate, we perform 10 different rollouts using this simulator with random initial states and compute the mean of the cumulative rewards. We perform one round of evaluation of the policy (student or baseline) during training at a fixed frequency. We take the average value of the last 10 evaluation rounds during training and report the mean and the standard deviation of the score. Lastly, since we run multiple trials (for each configuration), we compute the mean and standard deviation of the results of the three random trials for the given configuration. In order to facilitate easier understanding and comparison of the results across datasets and environments, we adopt the normalized score computation[5] (Fu et al., 2020)

$$\text{normalized score} = 100 \times \frac{\text{score - score of random policy}}{\text{score of expert online policy - score of random policy}}.$$

## C.3 DETAILED ANALYSIS

We designed our experimental analysis to answer the following questions.

- How does the improvement offered by the transfer algorithm vary with the dataset difficulty? See Figures 7a and 8a.

- How does the improvement offered by the transfer algorithm vary with the number of offline features that are left out of the online setting? See Figures 7b and 8b.

- We want to study how much performance is lost by not using the offline features (for the baseline), and what part of that performance is recovered by using the transfer algorithm (Table 5).

**Dataset Difficulty:** From Figures 7a and 8a, we observe that majority of the performance gains have been observed for the medium or medium-replay datasets. The expert and medium-expert datasets consistently provided relatively lesser % improvement, and in some cases performed worse than the baseline. The trend is consistent across the different environments (although it is more significant for HalfCheetah-v2). With expert or medium-expert datasets, the coverage of the state-action product space is narrow but of good quality. When training with limited features, the trained policy can drift away from this distribution. Since the data distribution is narrow, it is difficult for the trained policy to distinguish between unsafe state-action pairs from the safe state-action pairs. Whereas for the datasets with poor quality (medium, medium-replay), the data distribution is wide. The regularization from the teacher is stronger and also the agent can better discriminate unsafe state-action pairs due to the good coverage of the data.

**Reduced Feature Dimension:** From Fig 8b, we observe that the performance gains with the proposed algorithm follows an inverted "U" pattern where the performance gain is highest when the reduced dimension is neither too less nor too large. When the reduced dimension is small, the online and offline features might be less correlated and hence the transfer is not efficient. When the online feature dimension is already high, the baseline has a decent performance and therefore the % performance improvement over the baseline is limited. This trend is more pronounced for the results on D4RL datasets but not very clear on the RC-D4RL datasets.

---

[3] 3 environments x 4 dimensions x 5 seeds x 4 difficulty levels.

[4] The expert data collection policy affects the quality of all the 4 difficulties.

[5] Expert online policy is a fully trained online SAC policy (Haarnoja et al., 2018)

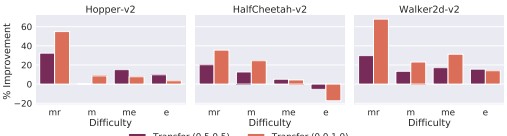

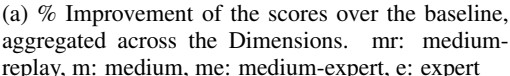

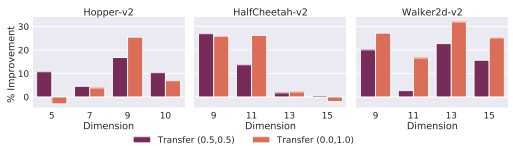

(a) % Improvement of the scores over the baseline, aggregated across the Dimensions. mr: medium-replay, m: medium, me: medium-expert, e: expert

(b) % Improvement of the scores over the baseline, aggregated across the difficulties.

Figure 7: Results on RC-D4RL datasets.

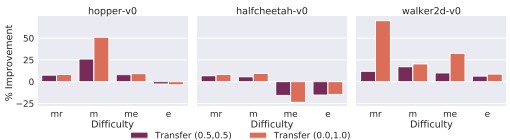

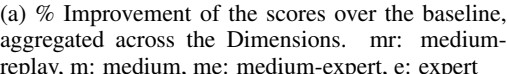

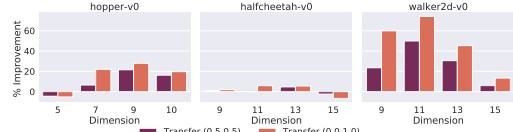

(a) % Improvement of the scores over the baseline, aggregated across the Dimensions. mr: medium-replay, m: medium, me: medium-expert, e: expert

(b) % Improvement of the scores over the baseline, aggregated across the difficulties.

Figure 8: Results on D4RL datasets.

| Difficulty | Dimension | Baseline | Transfer (0.5,0.5) | Transfer (0.0,1.0) | % Recovered (0.5,0.5) | % Recovered (0.0,1.0) |
|---|---|---|---|---|---|---|
| medium-replay | 9 | -36.0 % | -25.7 % | -17.6 % | 10.3 % | 18.4 % |
| medium-replay | 11 | -40.3 % | -28.8 % | -18.0 % | 11.5 % | 22.3 % |
| medium-replay | 13 | -35.3 % | -27.8 % | -19.3 % | 7.5 % | 16.0 % |
| medium-replay | 15 | -21.5 % | -7.3 % | -1.6 % | 14.1 % | 19.9 % |
| medium | 9 | -77.9 % | -63.4 % | -56.6 % | 14.5 % | 21.4 % |
| medium | 11 | -67.1 % | -66.4 % | -60.5 % | 0.7 % | 6.6 % |
| medium | 13 | -37.2 % | -36.6 % | -30.1 % | 0.5 % | 7.1 % |
| medium | 15 | -12.4 % | -5.7 % | -4.3 % | 6.7 % | 8.1 % |
| medium-expert | 9 | -43.2 % | -31.8 % | -37.5 % | 11.4 % | 5.7 % |
| medium-expert | 11 | -43.6 % | -36.4 % | -33.3 % | 7.3 % | 10.3 % |
| medium-expert | 13 | -26.3 % | -19.2 % | -16.1 % | 7.1 % | 10.2 % |
| medium-expert | 15 | -3.6 % | -10.4 % | -12.3 % | -6.8 % | -8.8 % |
| expert | 9 | -56.5 % | -56.5 % | -64.3 % | 0.0 % | -7.7 % |
| expert | 11 | -41.1 % | -39.3 % | -39.3 % | 1.8 % | 1.8 % |
| expert | 13 | -15.5 % | -23.9 % | -35.6 % | -8.5 % | -20.1 % |
| expert | 15 | -6.3 % | -10.6 % | -19.2 % | -4.3 % | -12.9 % |

Table 5: We summarize the loss in performance by not using the offline features for the Baseline, Transfer (0.5,0.5) and Transfer (0.0,1.0) as a percentage change over the Teacher score. We also show the % of recovered performance by using the Transfer algorithm as compared to the Baseline in the last two columns.

**% of Teacher's performance:** Here, we present the analysis of the drop in performance by using only online features (in the baseline) by measuring the percentage difference as compared to the teacher model (trained using full offline features). As we can observe, the baseline consistently under performs the teacher (as far as -70% in some cases). We also show the percentage drop in performance of the proposed transfer method trained using the teacher and that the transfer approach can recover some lost performance by not using the offline features. Interestingly, we observe that for higher quality datasets, the improvement offered by the transfer approach is limited. Moreover, we observe that increasing the weight of the teacher has a positive impact on the performance (as can also be seen for the Ads results and Atari results).

| Difficulty | Dimension | True-BC | Predictive | Transfer (0.0,1.0) |
|---|---|---|---|---|
| medium-replay | 9 | 8.1 | 8.1 | **11.2** |
| medium-replay | 11 | 8.1 | 8.4 | **11.4** |
| medium-replay | 13 | 13.9 | 12.1 | **15.1** |
| medium-replay | 15 | 13.9 | 15.0 | **18.8** |
| expert | 9 | **13.1** | 9.9 | 6.0 |
| expert | 11 | **14.4** | 9.9 | 9.9 |
| expert | 13 | **24.8** | 24.2 | 18.3 |
| expert | 15 | **34.0** | 31.5 | 28.2 |

Table 6: Performance of True-BC and Predictive baseline on RC-D4RL HalfCheetah-v2 datasets

## C.4 ADDITIONAL BASELINES

### C.4.1 PURE BEHAVIOR CLONING TRANSFER (FROM TEACHER)

In addition to the TD3+BC baseline that we consider to evaluate in the online features case, we also evaluate a pure behavior cloning algorithm that takes a teacher policy as input and learns to imitate the teacher. The new policy is learnt by minimizing

$$\arg\min_{\phi} \mathbf{E}_{s_i \sim D}\left[\left(\pi_{\text{teacher}}(s_i) - \pi_\phi(\hat{s}_i)\right)^2\right].$$

The policy $\pi_\phi$ uses a similar architecture as the teacher, with the exception that the input number of features is reduced due to the online features available.

We evaluate the algorithm on HalfCheetah environment in the RC-D4RL datasets, and summarize the results in the following table. The training and evaluation procedure is similar as the main experiments.

### C.4.2 PREDICTIVE MODEL

We also consider an additional baseline that predicts the missing features (offline features) from the available online features. We do this by first training an autoencoder that takes the online features as input and predicts the offline features by minimizing the MSE loss between the predicted offline features and the actual offline features. The trained autoencoder is than passed to the offline RL algorithm (that is trained for deployment). During every step of training, the algorithm takes the online features, predicts the offline features using the autoencoder and uses the predicted features as the state observation. Similarly, during evaluation, the trained agent first predicts the features using online features and uses them to take an action.

We evaluate the algorithm on HalfCheetah environment in the RC-D4RL datasets, and summarize the results in the following table. The training and evaluation procedure is similar as the main experiments.

### C.4.3 RESULTS

We can observe from the results that the True BC agent is very effective in the expert dataset (which is of high quality), whereas the proposed algorithm is effective in medium-replay dataset (which is of low quality). Similar conclusions hold true for the predictive baseline (autoencoder). Although the performance on the expert datasets is quite good for these baselines, real world datasets are often a mixture of datasets from multiple policies and are often noisy. Although these algorithms provide a strong starting point, they may not be quite useful in real applications.

## C.5 LEARNING CURVES

Figures 9, 10, 11 depict the learning curves during training in the RC-D4RL experiments.

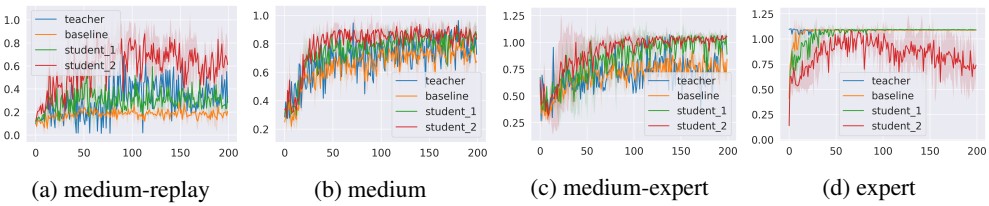

| (a) medium-replay | (b) medium | (c) medium-expert | (d) expert |

Figure 9: Hopper

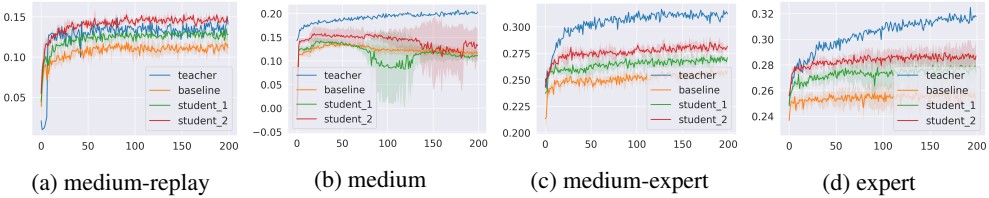

| (a) medium-replay | (b) medium | (c) medium-expert | (d) expert |

Figure 10: HalfCheetah

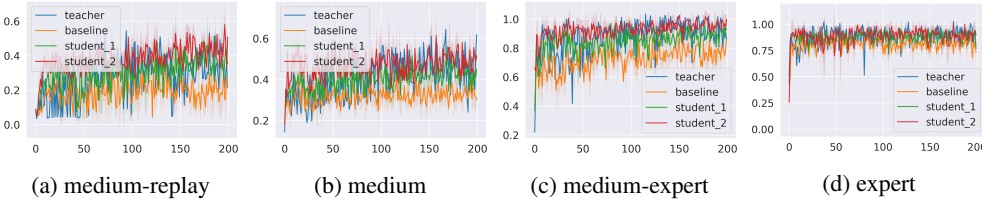

| (a) medium-replay | (b) medium | (c) medium-expert | (d) expert |

Figure 11: Walker2d

## C.6 DATASET SUMMARY

We present the summary of the RC-D4RL datasets.

| Dataset | Reduced Dim: 5 | 7 | 9 | 10 |
|---|---|---|---|---|
| medium-replay | $47.4 \pm 23.4$ | $83.3 \pm 41.0$ | $186.0 \pm 76.1$ | $207.3 \pm 45.5$ |
| medium | $588.1 \pm 351.5$ | $754.4 \pm 339.0$ | $900.0 \pm 285.5$ | $1357.8 \pm 423.6$ |
| medium-expert | $660.6 \pm 335.7$ | $936.7 \pm 395.1$ | $1380.9 \pm 473.1$ | $1929.7 \pm 469.3$ |
| expert | $803.6 \pm 320.0$ | $1196.9 \pm 402.8$ | $2399.0 \pm 744.9$ | $3068.5 \pm 617.1$ |

Table 7: Hopper environment: Average reward per trajectory

| Dataset | Reduced Dim: 9 | 11 | 13 | 15 |
|---|---|---|---|---|
| medium-replay | $-77.5 \pm 140.9$ | $-31.6 \pm 141.2$ | $183.7 \pm 288.9$ | $206.2 \pm 276.5$ |
| medium | $612.8 \pm 321.5$ | $838.3 \pm 531.1$ | $1389.1 \pm 637.9$ | $1786.6 \pm 931.7$ |
| medium-expert | $1013.2 \pm 519.6$ | $1450.2 \pm 873.8$ | $2471.6 \pm 1073.3$ | $3404.9 \pm 1803.1$ |
| expert | $1413.5 \pm 724.8$ | $2062.2 \pm 1233.0$ | $3554.1 \pm 1528.2$ | $5023.2 \pm 2682.1$ |

Table 8: HalfCheetah environment: Average reward per trajectory

| Dataset | Reduced Dim: 9 | 11 | 13 | 15 |
|---|---|---|---|---|
| medium-replay | $19.8 \pm 17.6$ | $43.2 \pm 65.2$ | $175.2 \pm 131.6$ | $269.7 \pm 73.2$ |
| medium | $307.2 \pm 106.2$ | $446.8 \pm 182.9$ | $797.8 \pm 208.4$ | $1071.0 \pm 188.5$ |
| medium-expert | $424.1 \pm 128.8$ | $650.0 \pm 273.4$ | $1173.3 \pm 367.4$ | $1743.9 \pm 341.4$ |
| expert | $691.9 \pm 302.2$ | $1039.5 \pm 507.4$ | $1837.0 \pm 735.4$ | $3330.7 \pm 685.7$ |

Table 9: Walker2d environment: Average reward per trajectory

## C.7 MORE ANALYSIS

Figures 12, 13, 14 illustrate the results of our experiments with RC-D4RL datsets. We plot the mean normalized score and the error bars represent the standard deviation across the random seeds.

Tables 10, 11, 12 contain the results of the RC-D4RL experiments. 13, 14, 15 contain the results of the D4RL experiments. In both cases, we report the mean normalized score and the standard deviation.

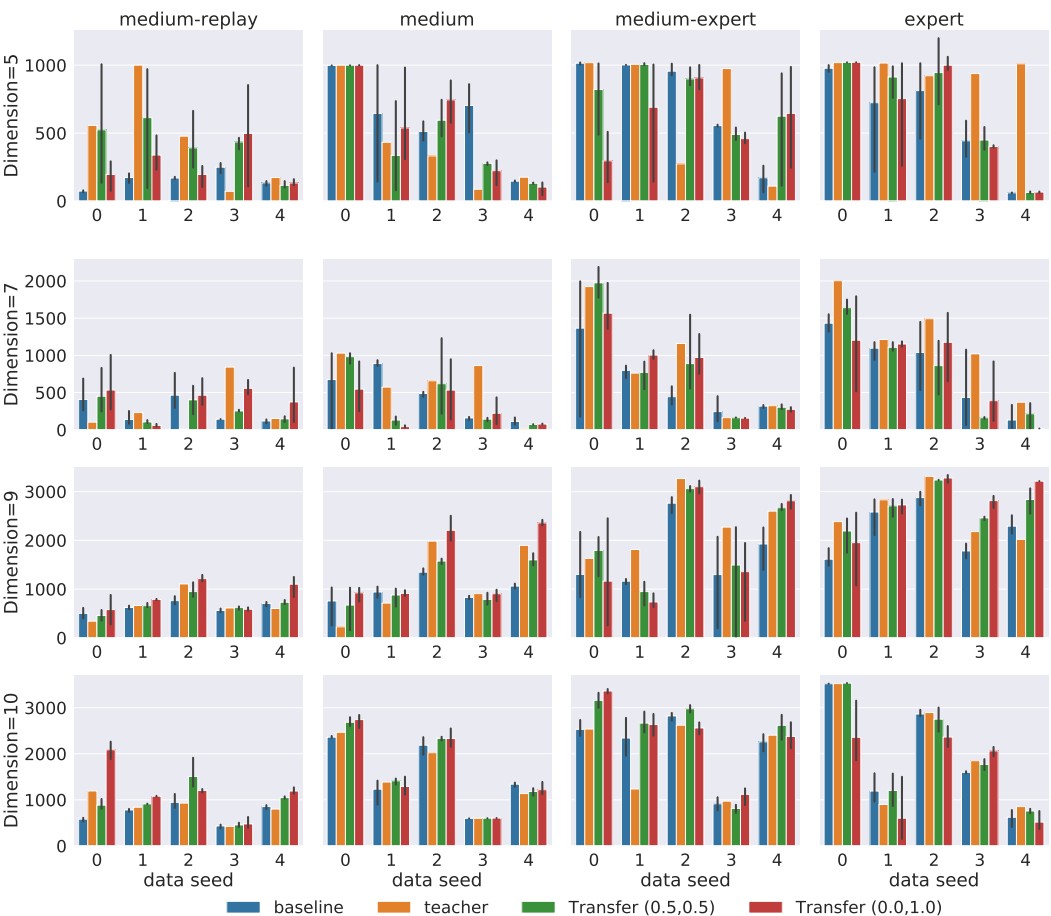

Figure 12: RC-D4RL Hopper-v2 experiments summary. We plot the mean of the rewards and the error bars represent the standard deviation across the 3 random seeds for a given dataset.

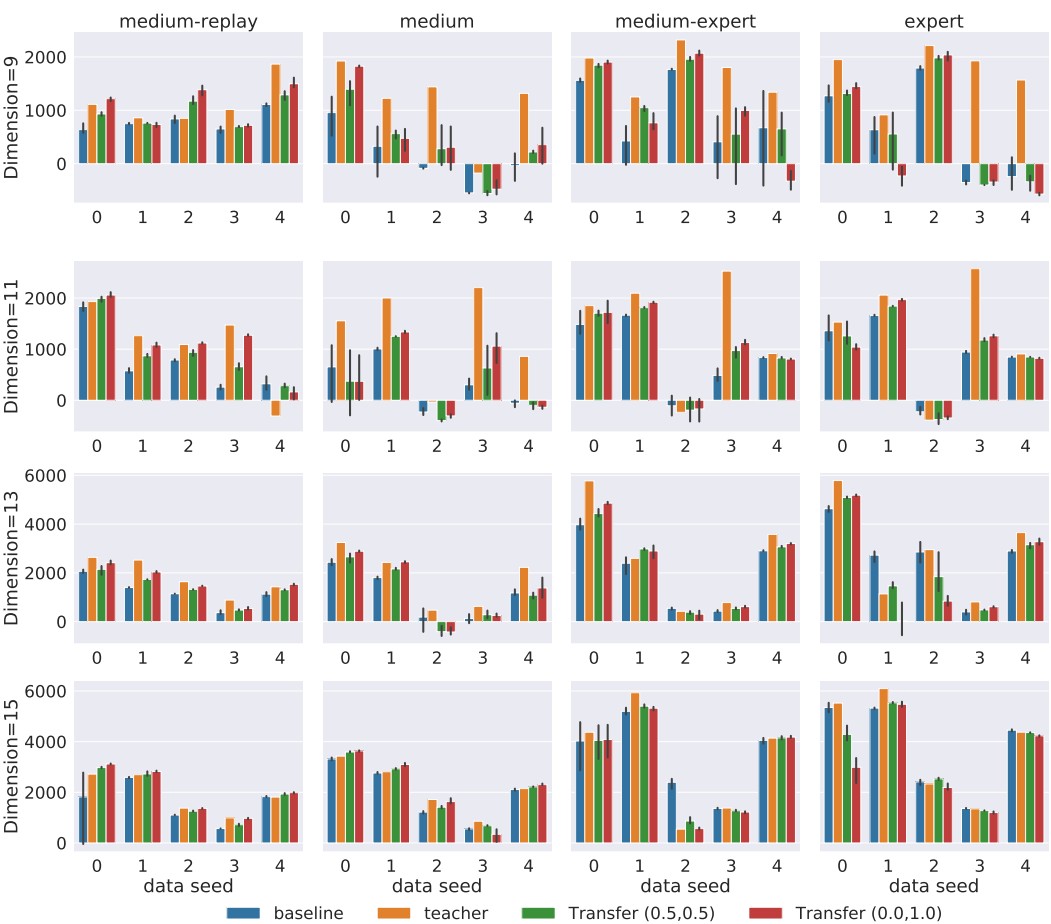

Figure 13: RC-D4RL HalfCheetah-v2 experiments summary. We plot the mean of the rewards and the error bars represent the standard deviation across the 3 random seeds for a given dataset.

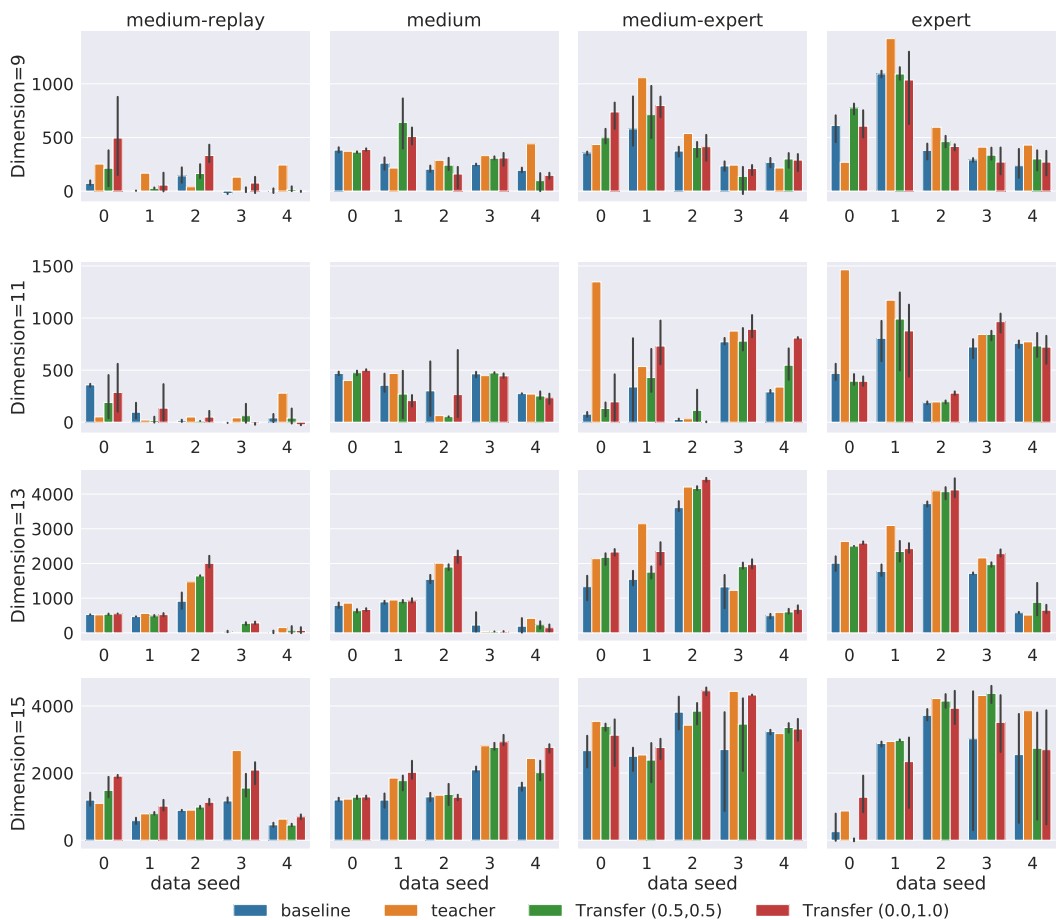

Figure 14: RC-D4RL Walker2d-v2 experiments summary. We plot the mean of the rewards and the error bars represent the standard deviation across the 3 random seeds for a given dataset.

| Difficulty | Dimension | Baseline | Transfer (0.5,0.5) | Transfer (0.0,1.0) |
|---|---|---|---|---|
| medium-replay | 5 | $5.5 \pm 1.9$ | **13.4** $\pm 9.6$ | $9.0 \pm 6.4$ |
| | 7 | $8.4 \pm 6.4$ | $9.0 \pm 6.2$ | **12.9** $\pm 9.2$ |
| | 9 | $20.0 \pm 3.6$ | $21.6 \pm 5.6$ | **26.9** $\pm 9.5$ |
| | 10 | $22.7 \pm 6.3$ | $30.2 \pm 11.6$ | **37.7** $\pm 16.6$ |
| medium | 5 | **19.1** $\pm 10.5$ | $15.0 \pm 10.7$ | $16.7 \pm 11.6$ |
| | 7 | **14.9** $\pm 11.6$ | $12.6 \pm 13.0$ | $9.4 \pm 9.5$ |
| | 9 | $31.0 \pm 8.5$ | $34.5 \pm 14.2$ | **45.5** $\pm 21.8$ |
| | 10 | $48.0 \pm 21.1$ | **51.1** $\pm 24.4$ | $50.9 \pm 25.3$ |
| medium-expert | 5 | $23.3 \pm 10.6$ | **24.2** $\pm 8.6$ | $19.0 \pm 10.0$ |
| | 7 | $20.1 \pm 18.0$ | **25.8** $\pm 21.6$ | $25.1 \pm 17.3$ |
| | 9 | $52.5 \pm 24.6$ | **61.9** $\pm 29.4$ | $57.0 \pm 34.5$ |
| | 10 | $67.4 \pm 21.8$ | **75.8** $\pm 27.2$ | $74.7 \pm 23.8$ |
| expert | 5 | $19.2 \pm 12.1$ | **21.5** $\pm 12.2$ | $20.5 \pm 12.7$ |
| | 7 | **26.0** $\pm 17.5$ | $25.1 \pm 18.4$ | $24.8 \pm 19.0$ |
| | 9 | $69.1 \pm 16.3$ | $83.2 \pm 12.7$ | **86.6** $\pm 17.8$ |
| | 10 | $60.8 \pm 34.5$ | **62.2** $\pm 32.7$ | $49.2 \pm 29.6$ |

Table 10: Results on RC-D4RL Hopper-v2 dataset

| Difficulty | Dimension | Baseline | Transfer (0.5,0.5) | Transfer (0.0,1.0) |
|---|---|---|---|---|
| medium-replay | 9 | $8.7 \pm 1.5$ | $10.1 \pm 2.0$ | **11.2** $\pm 2.8$ |
| | 11 | $8.3 \pm 4.8$ | $9.9 \pm 4.8$ | **11.4** $\pm 5.0$ |
| | 13 | $12.1 \pm 4.6$ | $13.5 \pm 4.7$ | **15.1** $\pm 5.3$ |
| | 15 | $15.0 \pm 7.6$ | $17.7 \pm 7.1$ | **18.8** $\pm 6.9$ |
| medium | 9 | $3.2 \pm 4.7$ | $5.3 \pm 5.4$ | **6.3** $\pm 6.5$ |
| | 11 | $5.0 \pm 4.2$ | $5.1 \pm 5.4$ | **6.0** $\pm 5.6$ |
| | 13 | $11.5 \pm 7.7$ | $11.6 \pm 9.6$ | **12.8** $\pm 10.6$ |
| | 15 | $18.3 \pm 8.4$ | $19.7 \pm 8.7$ | **20.0** $\pm 9.7$ |
| medium-expert | 9 | $10.0 \pm 6.1$ | **12.0** $\pm 5.7$ | $11.0 \pm 7.3$ |
| | 11 | $9.3 \pm 5.5$ | $10.5 \pm 6.1$ | **11.0** $\pm 6.3$ |
| | 13 | $18.8 \pm 11.5$ | $20.6 \pm 13.1$ | **21.4** $\pm 14.2$ |
| | 15 | **29.7** $\pm 11.7$ | $27.6 \pm 14.9$ | $27.0 \pm 15.5$ |
| expert | 9 | **7.3** $\pm 7.2$ | **7.3** $\pm 8.0$ | $6.0 \pm 8.9$ |
| | 11 | $9.6 \pm 5.4$ | **9.9** $\pm 6.2$ | **9.9** $\pm 6.3$ |
| | 13 | **24.0** $\pm 11.4$ | $21.6 \pm 13.5$ | $18.3 \pm 16.4$ |
| | 15 | **32.7** $\pm 13.5$ | $31.2 \pm 12.6$ | $28.2 \pm 12.7$ |

Table 11: Results on RC-D4RL HalfCheetah-v2 dataset

| Difficulty | Dimension | Baseline | Transfer (0.5,0.5) | Transfer (0.0,1.0) |
|---|---|---|---|---|
| medium-replay | 9 | $0.8 \pm 1.5$ | $1.8 \pm 2.5$ | **4.1** $\pm 5.4$ |
| | 11 | **2.2** $\pm 3.1$ | $1.4 \pm 2.6$ | $1.9 \pm 3.7$ |
| | 13 | $8.5 \pm 7.8$ | $13.0 \pm 12.4$ | **14.9** $\pm 15.5$ |
| | 15 | $18.7 \pm 7.1$ | $22.9 \pm 10.3$ | **29.8** $\pm 12.6$ |
| medium | 9 | $5.6 \pm 1.6$ | **7.1** $\pm 4.6$ | $6.5 \pm 3.4$ |
| | 11 | **8.1** $\pm 3.0$ | $6.6 \pm 4.0$ | $7.2 \pm 4.1$ |
| | 13 | $15.8 \pm 11.7$ | $16.2 \pm 14.8$ | **17.5** $\pm 17.8$ |
| | 15 | $32.2 \pm 8.1$ | $40.0 \pm 12.8$ | **44.8** $\pm 16.2$ |
| medium-expert | 9 | $7.9 \pm 3.5$ | $8.9 \pm 5.1$ | **10.6** $\pm 5.7$ |
| | 11 | $6.5 \pm 6.9$ | $8.7 \pm 6.4$ | **11.4** $\pm 8.5$ |
| | 13 | $36.1 \pm 24.1$ | $46.2 \pm 26.1$ | **51.2** $\pm 27.4$ |
| | 15 | $65.0 \pm 18.3$ | $71.6 \pm 15.9$ | **78.3** $\pm 17.0$ |
| expert | 9 | $11.3 \pm 7.3$ | **12.9** $\pm 6.9$ | $11.3 \pm 7.3$ |
| | 11 | $12.8 \pm 5.6$ | $13.7 \pm 7.6$ | **14.1** $\pm 7.0$ |
| | 13 | $42.7 \pm 22.9$ | $51.2 \pm 23.7$ | **52.6** $\pm 25.0$ |
| | 15 | $54.1 \pm 36.3$ | **62.0** $\pm 38.4$ | $59.9 \pm 29.6$ |

Table 12: Results on RC-D4RL Walker2d-v2 dataset

| Difficulty | Dimension | Baseline | Transfer (0.5,0.5) | Transfer (0.0,1.0) |
|---|---|---|---|---|
| medium-replay | 5 | $6.3 \pm 3.3$ | $8.0 \pm 4.2$ | **8.2** $\pm 4.4$ |
| | 7 | $14.7 \pm 4.5$ | $19.0 \pm 4.9$ | **20.3** $\pm 6.9$ |
| | 9 | $29.3 \pm 3.0$ | $30.6 \pm 2.1$ | **30.9** $\pm 2.7$ |
| | 10 | $31.5 \pm 1.0$ | **32.3** $\pm 1.2$ | $31.1 \pm 0.7$ |
| medium | 5 | **10.6** $\pm 3.6$ | $8.4 \pm 2.3$ | $6.5 \pm 1.7$ |
| | 7 | $29.6 \pm 7.0$ | $35.5 \pm 19.1$ | **50.4** $\pm 30.7$ |
| | 9 | $43.4 \pm 12.7$ | $73.0 \pm 17.1$ | **88.6** $\pm 18.5$ |
| | 10 | $66.9 \pm 28.7$ | $87.6 \pm 18.3$ | **99.6** $\pm 0.5$ |
| medium-expert | 5 | $6.6 \pm 2.8$ | $7.5 \pm 3.4$ | **8.2** $\pm 3.0$ |
| | 7 | $29.2 \pm 15.4$ | $35.5 \pm 22.0$ | **43.0** $\pm 26.5$ |
| | 9 | $85.6 \pm 29.0$ | **104.6** $\pm 7.3$ | $101.2 \pm 5.3$ |
| | 10 | $107.9 \pm 3.3$ | **111.0** $\pm 1.3$ | $109.0 \pm 2.2$ |
| expert | 5 | **9.1** $\pm 2.2$ | $7.4 \pm 0.7$ | $8.1 \pm 1.6$ |
| | 7 | **46.9** $\pm 33.6$ | $38.1 \pm 37.6$ | $33.1 \pm 33.3$ |
| | 9 | **61.5** $\pm 50.6$ | $58.9 \pm 42.5$ | $60.4 \pm 39.7$ |
| | 10 | $70.8 \pm 55.5$ | $91.0 \pm 46.6$ | **92.0** $\pm 44.4$ |

Table 13: Results on D4RL Hopper-v0 dataset

| Difficulty | Dimension | Baseline | Transfer (0.5,0.5) | Transfer (0.0,1.0) |
|---|---|---|---|---|
| medium-replay | 9 | $26.5 \pm 5.8$ | $29.4 \pm 5.8$ | **30.0** $\pm 6.3$ |
| | 11 | $33.1 \pm 5.6$ | $35.4 \pm 4.5$ | **36.1** $\pm 4.4$ |
| | 13 | $32.0 \pm 8.9$ | **35.7** $\pm 6.8$ | $35.6 \pm 6.7$ |
| | 15 | $38.9 \pm 2.0$ | $41.0 \pm 1.2$ | **41.6** $\pm 1.3$ |
| medium | 9 | $33.5 \pm 4.1$ | $35.6 \pm 3.8$ | **37.3** $\pm 3.5$ |
| | 11 | $36.1 \pm 2.5$ | $39.1 \pm 0.9$ | **40.5** $\pm 1.2$ |
| | 13 | $36.9 \pm 2.0$ | $39.3 \pm 1.4$ | **40.8** $\pm 1.2$ |
| | 15 | $40.2 \pm 1.2$ | $41.5 \pm 1.0$ | **42.5** $\pm 0.7$ |
| medium-expert | 9 | **14.7** $\pm 8.6$ | $13.1 \pm 8.0$ | $11.9 \pm 5.9$ |
| | 11 | **21.4** $\pm 5.5$ | $19.6 \pm 4.9$ | $18.7 \pm 2.7$ |
| | 13 | **25.3** $\pm 14.9$ | $25.3 \pm 13.8$ | $25.3 \pm 14.8$ |
| | 15 | **40.8** $\pm 8.4$ | $37.9 \pm 12.9$ | $31.1 \pm 6.4$ |
| expert | 9 | $6.2 \pm 2.0$ | $6.6 \pm 2.8$ | **6.7** $\pm 2.9$ |
| | 11 | $8.8 \pm 6.5$ | $9.1 \pm 7.6$ | **12.8** $\pm 10.5$ |
| | 13 | $24.5 \pm 25.7$ | **25.0** $\pm 29.8$ | $24.7 \pm 28.8$ |
| | 15 | **59.4** $\pm 22.1$ | $55.4 \pm 25.0$ | $52.8 \pm 23.0$ |

Table 14: Results on D4RL HalfCheetah-v0 dataset

| Difficulty | Dimension | Baseline | Transfer (0.5,0.5) | Transfer (0.0,1.0) |
|---|---|---|---|---|
| medium-replay | 9 | $2.3 \pm 2.0$ | $3.7 \pm 4.7$ | **7.4** $\pm 3.6$ |
| | 11 | $3.0 \pm 4.4$ | $5.2 \pm 4.4$ | **8.3** $\pm 4.3$ |
| | 13 | $6.9 \pm 1.6$ | $10.2 \pm 3.4$ | **12.5** $\pm 3.5$ |
| | 15 | $11.2 \pm 2.7$ | $10.0 \pm 5.8$ | **16.0** $\pm 3.7$ |
| medium | 9 | $8.8 \pm 5.5$ | **11.5** $\pm 5.5$ | $10.7 \pm 4.2$ |
| | 11 | $12.8 \pm 6.0$ | **20.3** $\pm 8.9$ | $17.1 \pm 8.3$ |
| | 13 | $25.9 \pm 13.1$ | **37.0** $\pm 14.1$ | $35.1 \pm 20.4$ |
| | 15 | $50.5 \pm 12.8$ | $57.9 \pm 15.1$ | **67.3** $\pm 12.4$ |
| medium-expert | 9 | $7.1 \pm 4.1$ | $10.1 \pm 6.2$ | **15.4** $\pm 9.9$ |
| | 11 | $15.8 \pm 8.0$ | $24.6 \pm 5.1$ | **37.5** $\pm 20.7$ |
| | 13 | $39.5 \pm 17.4$ | $50.6 \pm 27.8$ | **67.4** $\pm 14.7$ |
| | 15 | $89.1 \pm 9.6$ | $95.1 \pm 14.7$ | **96.9** $\pm 10.4$ |
| expert | 9 | $11.3 \pm 8.3$ | $11.1 \pm 6.7$ | **13.8** $\pm 8.1$ |
| | 11 | $21.9 \pm 8.6$ | $30.1 \pm 10.5$ | **30.2** $\pm 13.3$ |
| | 13 | $48.9 \pm 27.3$ | $60.4 \pm 25.1$ | **61.1** $\pm 28.8$ |
| | 15 | $93.4 \pm 8.3$ | $95.6 \pm 9.1$ | **96.5** $\pm 7.8$ |

Table 15: Results on D4RL Walker2d-v0 dataset

