# OpenReview forum: "Offline Reinforcement Learning with Resource Constrained Online Deployment"
_ICLR.cc/2022/Conference — ICLR 2022 Submitted_

### Official Review · Reviewer_oGRd · 2021-10-27

**Correctness:** 4
**Technical Novelty And Significance:** 1
**Empirical Novelty And Significance:** 3
**Recommendation:** 5
**Confidence:** 2

**Main Review:**

Motivation & significance: this paper introduces an interesting and novel offline RL setting and considers a simple modification to an existing algorithm for improved performance. It lists two real applications as the motivating examples. While it seems to be the real problem for the two examples, I doubt how general the proposed setting will applly in practice, and therefore how broadly the research in this setting will benefit to the RL community. The authors do carry out extensive experiments on the three environments from D4RL, but I am disappointed that no experiments are done for any real application or a simulated setting that resembles some property of either of the motivating examples.

Novelty: the proposed solution is quite simple and straightforward, adding an additional regularisation to a teacher policy with a higher quality. Therefore, there is very limited contribution to the methodology.

Clarity: the idea and experiments are very well presented in the paper. The algorithm design choice and experiment settings are explained in details and it is easy for reproducing the results.

Empirical evaluation: the experiments show the proposed method (both variants) is robust and provide improvement in most situations. I appreciate the authors makes a good effort to consider different experiment settings to get a comprehensive study about the proposed method and compare it to the baseline. However, I’m concerned that all the experiments are restricted to three simple simulated environments in D4RL and the results may not provide much evidence to how they will generalise to other environments or any real problems. Also, a very simple baseline is missing that trains a pure behaviour cloning policy to learn the teacher policy.

Overall, I think it would make this work much more impactful if the authors could consider a real problem / simulated problem similar to a real problem, and study different approaches and show what the best performance gain we could obtain against an existing solution.

**Summary Of The Paper:**

This paper considers a novel offline RL setting with resource-constrained online deployment. In that setting, the agent has access to less information about the state in the online deployment phase than the available information from the offline dataset. It proposes a two stage training strategy under this setting, first training a teacher agent with full features, and then training a student agent with regularisation to mimic the teacher agent. Experiments are carried on in three D4RL environments with a variety configurations in the data collection protocol, behaviour policy, feature constraints. Results show the advantage of the proposed distilled policy than a simple baseline.

**Summary Of The Review:**

The new problem setting could be better motivated and supported by experiments. The proposed algorithm is a simple modification to the existing method.

---

> ### Author Response · Authors · 2021-11-21
> **Response to reviewer oGRd**
>
> ## Pure Behavior Cloning Transfer (From Teacher)
>
> We thank the reviewer for suggesting this additional baseline. Results are forthcoming and we will share them as they become available. We will also include these in the paper as space allows.
>
> Please also review the general response at your leisure, as it contains experiments on a real-life Ads dataset. We hope that these results alleviate your concerns about the applicability of the proposed setup/approach for practical problems.

---

> ### Author Response · Authors · 2021-11-23
> **Response to  Reviewer oGRd**
>
> We thank the reviewer for the constructive feedback.
>
> > While it seems to be the real problem for the two examples, I doubt how general the proposed setting will applly in practice, and therefore how broadly the research in this setting will benefit to the RL community. The authors do carry out extensive experiments on the three environments from D4RL, but I am disappointed that no experiments are done for any real application or a simulated setting that resembles some property of either of the motivating examples.
>
> We added additional experimental results on two domains: Atari 2600 and a real life Ads autobidding task.
> In the Atari 2600 experiments, we simulated a power constrained setting (as in the nano-satellite example) by restricting the student model to use an architecture with smaller number of parameters. The experiments on Ads data simulate the system latency setting since the high quality embeddings are difficult to procure online and also require larger models. We hope that we addressed your concerns about the experimental setup with the new results.
>
>
> > I’m concerned that all the experiments are restricted to three simple simulated environments in D4RL and the results may not provide much evidence to how they will generalise to other environments or any real problems
>
> As noted above, we included experiments on a diverse set of tasks to show the applicability of the work.
>
>
> > Also, a very simple baseline is missing that trains a pure behaviour cloning policy to learn the teacher policy.
>
> We thank you for suggesting to us this baseline. We evaluated this baseline for the HalfCheetah-v2 RC-D4RL datasets, and observed that it performs very strongly on the expert dataset which is of a high quality. On the medium-replay dataset however, the proposed transfer approach outperforms the True-BC baseline suggesting the utility of the proposed approach on lower quality datasets that are often the case in real world.
>
> | Difficulty          |   Dimension |   True-bc |   Autoencoder |   Transfer (0.0,1.0) |
> |:--------------|------:|---------------:|-------------------:|--------------------------:|
> | medium-replay |     9 |            8.1 |                8.1 |                      **11.2** |
> | medium-replay |    11 |            8.1 |                8.4 |                      **11.4** |
> | medium-replay |    13 |           13.9 |               12.1 |                      **15.1** |
> | medium-replay |    15 |           13.9 |               15   |                      **18.8** |
> | expert        |     9 |           **13.1** |                9.9 |                       6   |
> | expert        |    11 |           **14.4** |                9.9 |                       9.9 |
> | expert        |    13 |           **24.8** |               24.2 |                      18.3 |
> | expert        |    15 |           **34**   |               31.5 |                      28.2 |
>
>
>
> Please also review the overall responses above containing clarifying analyses that were requested by multiple reviewers. We hope that these results alleviate your concerns about the applicability of the proposed setup/approach for practical problems.

---

> > ### Comment · Reviewer_oGRd · 2021-11-26
> > **Thank you for the major revision**
> >
> > Thanks for providing a major revision with an extensive batch of new experiment results and a few baseline methods.
> >
> > My concern about the lack of motivation from real application is addressed with the new Ad experiment. The additional Atari experiments are also helpful to provide a different RL environment from the original settings.
> >
> > However, new concerns arise when reading through the other reviewers' comment, your responses and the update paper.
> >
> > 1. The experiment results are mixed. The advantage of the proposed method is not consistent in all settings, compared to simple baselines. Particularly, the simple True-BC agent that learns from the teacher policy does better in the expert setting while worse in the medium-replay setting of HalfCheetah. This result suggests more extensive comparison should be done in other experiments too. Also, the Transfer agent is not significantly or consistently better than the CQL baseline in Atari. It casts the doubt whether the proposed method is indeed a better choice than other simpler methods. In the discrete action setting including Atari and Ads, I think the True-BC agent can be trained as well, e.g., predicting the action directly or learning the Q value of a teacher's.
> >
> > 2. There are effectively two proposed transfer algorithms, one for continuous control, and the other for discrete actions. While the high level idea of using teacher's learned policy is shared, the detailed algorithm implementations are clearly different. I wonder what conclusion can we draw from this paper.
> >
> > 3. Reviewer CV2Z made a good point that this setting is very similar to sim2real and other works in online RL settings. Since the proposed transfer method is not specific to an offline RL algorithm, I suspect many ideas from those sim2real and online RL literatures could be borrowed directly to the problem setting of this paper as well.
> >
> > 4. As a minor point, the FQE evaluation in Figure 4 shows a significantly better performance in (0, 1) setting, and the results are almost the same as other parameter choices. I'm worried that this evaluation protocol could be confounded by the fact that the FQE algorithm is run with the full offline features as well. It might prefer the fully transfer algorithm (0, 1) just because it's Q function is estimated in a similar way as the transfer algorithm rather than the transfer algorithm with (0, 1) is really better than other choices in the real environment.
> >
> > Given the improved motivation and additional experiments, I'm willing to raise my setting to weak reject, but I feel more comprehensive experimental comparison and more consistent introduction of the proposed algorithm should be done to make a solid contribution.

---

> > > ### Author Response · Authors · 2021-11-28
> > > **Thanks for the comments**
> > >
> > > We thank the reviewer for the comments and updating the score of the paper. However, we find it unfortunate that the reviewer still rates it a weak reject.
> > >
> > > > results are mixed...
> > >
> > > We do agree that there are cases where the True-BC agent outperforms the proposed algorithm. However, the gap between the teacher and this baseline shows that the new problem setting we proposed in this paper is hard in addition to being important for practical applications. We do agree that more experiments need to be conducted, and are pursuing this currently. However, we wish the reviewer appreciates our contribution in proposing the novel problem setting (that is more practical than the sim2real setting where building a simulator for real world problems is very hard), the novel dataset collection and demonstrating the difficulty of the problem.
> > >
> > > > sim2real
> > >
> > > We agree that there could be more ways to solve the novel problem we proposed. We agree that some techniques from sim2real, transfer learning or distillation could also be useful. We want to highlight one point however. Sim2real line of work assumes that one can build a reasonable simulator of the environment which is not true for most real world systems.
> > > - consider the example in Ads or E-commerce in general: Several factors such as non-stationarity of user behavior, multi-agent behavior make it very challenging to build a simulator
> > > - Even in physical systems where dynamics of the model are studied, it is a very costly affair to build a high fidelity simulator
> > >
> > > We want to highlight that without having access to a good simulator, the teacher model is less perfect which is the setting in our case.
> > >
> > > We note that we will take the advice and conduct more research along this direction. We will implement some sim2real inspired baselines and will highly appreciate any method recommendations.
> > >
> > > > FQE
> > >
> > > We would like to clarify that the FQE for the agents (trained using online features) is also performed using the available online features and not the full offline features.

---

### Official Review · Reviewer_QjjH · 2021-11-01

**Correctness:** 3
**Technical Novelty And Significance:** 4
**Empirical Novelty And Significance:** 4
**Recommendation:** 5
**Confidence:** 4

**Main Review:**

**Strengths**


**S1:** The paper outlines a previously under-explored and an important class resource constrained setup for offline RL. Furthermore they define the data collection strategy inline with this setup which was not present in previously explored / popular datasets such as D4RL.


**S2:** They propose a minimal extension on TD3 + BC algorithm for the resource constrained settings and show that the proposed algorithm was able to close the performance gap due to the missing privileged information (I.e. the additional offline feature set).



**S3:** With a few exceptions (discussed below), The experiments presented are well designed.




**Weaknesses:**

**W1:** The motivation behind the RC-D4RL dataset is not well articulated. As articulated in the paper, the available dataset D4RL is collected by a policy that has access to the privileged information, which in turn results in the behavioral policy having high quality trajectories. These high quality action choices should not be expected form a policy trained with no access to the privileged information. Hence it is important to collect the dataset with a resource constrained policy (no access to privileged information) in order to better quantify the advantages of being able to leverage privileged information.

**Q:** While the proposed setting makes sense in imitating real world settings, would it not be useful to have high quality trajectories in order to best leverage the action choices that come from these high quality offline features. Why not stick with D4RL?
**Q:** And if it indeed the case that D4RL does not represent real world settings why are we evaluating the given algorithm for both RC-D4RL and D4RL settings?





**W2:** The paper seems to be missing some natural baselines. While the proposed algorithm is quite straightforward and is able to beat a blind agent (blind to privileged information) it would be interesting to see how it compares to other natural baselines such as a predictive model baseline.



Predictive mode baseline: Train a model to predict the privileged information and use this in tandem with a offline RL model that is trained with the privileged information.



Also If I understand it correctly, the baseline would simply by equivalent to Transfer(1.0, 0). Then the question becomes did we just propose an algorithm and just compared the different hyperparameter settings?


**Q:** If a very natural baseline is comparable to the proposed algorithm, is the main contribution of the paper limited to the definition of the resource constrained framework?





**W3:** More Datasets please: While this may come off as a "knee-jerk" reaction, I do believe that the class of problem defined here is important for the community. Hence it would definitely help if we are able to find real world examples of such scenarios and incorporate them here. Some new more natural additions on the dataset may include, (1) Atari benchmark dataset with semantic maps generated from internal bits. (2) D4RL dataset coupled with RGB frames, (3) Self driving / navigation dataset coupled with depth/semantic maps.




**Clarity Issues/ Comments:**


**CL1:** While the paper concludes that the performance gap between the use of online features and offline features have been highlighted, it is not well quantified in the paper, how much does the overall performance suffer due to these missing offline features. I am guessing this would amount to teacher performances – baseline performance from fig 11. It would be nice to have them in the main paper.

**CL2:** When we say we address the performance gap, it would be nice to produce numbers that represent exactly that. I.e. What percentage of the performance gap was the addition of offline features able to fill.

**CL3:** The inverted U pattern mentioned for figures 4b and 5b are very hard to grasp, and basically nonexistent for 5b unless I am parsing the sentence incorrectly.
**CL4:** Figure 6 does not fully answer the question of how "important" the teacher's role is in training the student, were we expecting Transfer(0.5,0.5) to perform worse that Transfer(0, 1)? How do we maximize the teachers role in training without hurting performance?

**Summary Of The Paper:**

The authors explore Learning Using Privileged Information (LUPI) paradigm for offline reinforcement learning and explore a teacher-student framework for transferring a policy learned under privileged information. The proposed algorithm is a minimal extension of recently proposed algorithm TD3+BC to leverages the offline/priviliged information for online deployment.

**Summary Of The Review:**

The framework introduced by the authors represent an important part of offline RL that remains under explored. The proposed algorithm is a minimal extension to a popular offline RL and the experiments/results are well motivated. However, at the current state I would lean towards rejection as the paper still lacks detailed investigation of possible approaches to solve this problem as well as results on a more compelling set of benchmarks(datasets).

---

> ### Author Response · Authors · 2021-11-21
> **Response to revieiwer QjjH**
>
> > W1: The motivation behind the RC-D4RL dataset is not well articulated. As articulated in the paper, the available dataset D4RL is collected by a policy that has access to the privileged information, which in turn results in the behavioral policy having high quality trajectories. These high quality action choices should not be expected form a policy trained with no access to the privileged information. Hence it is important to collect the dataset with a resource constrained policy (no access to privileged information) in order to better quantify the advantages of being able to leverage privileged information.
>
> > Q: While the proposed setting makes sense in imitating real world settings, would it not be useful to have high quality trajectories in order to best leverage the action choices that come from these high quality offline features. Why not stick with D4RL?
> Q: And if it indeed the case that D4RL does not represent real world settings why are we evaluating the given algorithm for both RC-D4RL and D4RL settings?
>
> As the reviewer rightly points out, the setting of RC-D4RL imitates real-world settings, and is therefore more practical for evaluating the resource-constrained setting algorithms. It is true that having high quality trajectories would be more useful but it may not be practical in all offline settings. However, we show results on D4RL to emphasize that the proposed algorithm can indeed, when they are available, make use of 'high-quality trajectories' from data collected with policies that had access to privileged information.
>
>
>
> > W2: The paper seems to be missing some natural baselines. While the proposed algorithm is quite straightforward and is able to beat a blind agent (blind to privileged information) it would be interesting to see how it compares to other natural baselines such as a predictive model baseline. Predictive mode baseline: Train a model to predict the privileged information and use this in tandem with a offline RL model that is trained with the privileged information. Also If I understand it correctly, the baseline would simply by equivalent to Transfer(1.0, 0). Then the question becomes did we just propose an algorithm and just compared the different hyperparameter settings?
>
> > Q: If a very natural baseline is comparable to the proposed algorithm, is the main contribution of the paper limited to the definition of the resource constrained framework?
>
> We thank the reviewer for suggesting the predictive baseline. We are in the midst of using an autoencoder to implement the predictive baseline. Results are forthcoming, and we will share the results here as well as include them in the paper as space allows. Under our understanding, this baseline is not the same as Transfer(1.0,0).
>
> As the reviewer noted, in many experiments, we used Transfer (1.0,0) as our baseline. However, we want to highlight that Transfer (1.0,0) is actually the TD3+BC algorithm, which is a state-of-the-art offline RL baseline that we want to compare against our transfer algorithms (all settings of Transfer($\beta_1,\beta_2$) with positive $\beta_2$). We extended the TD3+BC algorithm by adding the transfer learning term (i.e., all settings with $\beta_2 > 0$) and then showed that this extension indeed improved model performance.
>
>
> For the predictive baseline, we are estimating the offline state from the online state and using the predicted state for the algorithm (TD3 +BC). We are not sure if these are the same definitions as those suggested by the reviewer and seek clarity here.
>
>
> > CL1: While the paper concludes that the performance gap between the use of online features and offline features have been highlighted, it is not well quantified in the paper, how much does the overall performance suffer due to these missing offline features. I am guessing this would amount to teacher performances – baseline performance from fig 11. It would be nice to have them in the main paper.
>
> > CL2: When we say we address the performance gap, it would be nice to produce numbers that represent exactly that. I.e. What percentage of the performance gap was the addition of offline features able to fill.
>
> We will include these statistics in the main document.
>
> > CL3: The inverted U pattern mentioned for figures 4b and 5b are very hard to grasp, and basically nonexistent for 5b unless I am parsing the sentence incorrectly.
>
> We will rewrite this for clarity. Forthcoming.
>
>
>
>
> > CL4: Figure 6 does not fully answer the question of how "important" the teacher's role is in training the student, were we expecting Transfer(0.5,0.5) to perform worse that Transfer(0, 1)? How do we maximize the teachers role in training without hurting performance?
>
> Forthcoming.
>
> We thank the reviewer for their detailed and constructive feedback. We will address all these points fully soon, and include them in the paper as space allows.
>
> Please also review the general response at your leisure, as it contains experiments on a real-life Ads dataset.

---

> > ### Author Response · Authors · 2021-11-23
> > **Response to Reviewer QjjH: Updated**
> >
> >
> >
> > > W2: The paper seems to be missing some natural baselines. While the proposed algorithm is quite straightforward and is able to beat a blind agent (blind to privileged information) it would be interesting to see how it compares to other natural baselines such as a predictive model baseline. Predictive mode baseline: Train a model to predict the privileged information and use this in tandem with a offline RL model that is trained with the privileged information. Also If I understand it correctly, the baseline would simply by equivalent to Transfer(1.0, 0). Then the question becomes did we just propose an algorithm and just compared the different hyperparameter settings?
> >
> > > Q: If a very natural baseline is comparable to the proposed algorithm, is the main contribution of the paper limited to the definition of the resource constrained framework?
> >
> > We thank the reviewer for suggesting the predictive baseline. We provided results for the predictive baseline using an autoencoder to predict the offline features. We evaluated it for HalfCheetah-v2 (RC-D4RL) datasets and observe that the proposed algorithm performs better on medium-replay dataset (low-quality), whereas the autoencoder performs better on higher quality dataset (expert). Real world datasets are often noisy and are of poorer quality, therefore, the proposed transfer approach offers better utility compared to the predictive baseline. Moreover, one drawback of using an autoencoder is during online operation, it increases the computation on the part of the agent.
> >
> >
> > | Difficulty          |   Dimension |   True-bc |   Autoencoder |   Transfer (0.0,1.0) |
> > |:--------------|------:|---------------:|-------------------:|--------------------------:|
> > | medium-replay |     9 |            8.1 |                8.1 |                      **11.2** |
> > | medium-replay |    11 |            8.1 |                8.4 |                      **11.4** |
> > | medium-replay |    13 |           13.9 |               12.1 |                      **15.1** |
> > | medium-replay |    15 |           13.9 |               15   |                      **18.8** |
> > | expert        |     9 |           **13.1** |                9.9 |                       6   |
> > | expert        |    11 |           **14.4** |                9.9 |                       9.9 |
> > | expert        |    13 |           **24.8** |               24.2 |                      18.3 |
> > | expert        |    15 |           **34**   |               31.5 |                      28.2 |
> >
> > As the reviewer noted, in many experiments, we used Transfer (1.0,0) as our baseline. However, we want to highlight that Transfer (1.0,0) is actually the TD3+BC algorithm, which is a state-of-the-art offline RL baseline that we want to compare against our transfer algorithms (all settings of Transfer($\beta_1,\beta_2$) with positive $\beta_2$). We extended the TD3+BC algorithm by adding the transfer learning term (i.e., all settings with $\beta_2 > 0$) and then showed that this extension indeed improved model performance.
> >
> >
> > For the predictive baseline, we are estimating the offline state from the online state and using the predicted state for the algorithm (TD3 +BC). We are not sure if these are the same definitions as those suggested by the reviewer and seek clarity here.

---

> > > ### Author Response · Authors · 2021-11-23
> > > **Response to Reviewer QjjH: continued**
> > >
> > >
> > > > CL1: While the paper concludes that the performance gap between the use of online features and offline features have been highlighted, it is not well quantified in the paper, how much does the overall performance suffer due to these missing offline features. I am guessing this would amount to teacher performances – baseline performance from fig 11. It would be nice to have them in the main paper.
> > >
> > > > CL2: When we say we address the performance gap, it would be nice to produce numbers that represent exactly that. I.e. What percentage of the performance gap was the addition of offline features able to fill.
> > >
> > > Thanks for suggesting this analysis. We performed this analysis and summarized the results here for HalfCheetah-v2 RC-D4RL datasets. More detailed results can be found from Table 5 in the Appendix of the paper. These results indicate that when the dataset quality is poor, the transfer learning approach offers a greater improvement (lesser drop compared to teacher). Despite this, the performance gap between the teacher and the transfer algorithm is still high for practical purposes, and we believe more trailored transfer learning approaches are required for the offline RL resource constrained setting.
> > >
> > > | diff          |   baseline-teacher % |   transfer(0.5,0.5)-teacher % |   transfer(0.0,1.0)-teacher % |   diff-percent (0.5,0.5) |   diff-percent (0.0,1.0) |
> > > |:--------------|---------------------:|------------------------------:|------------------------------:|-------------------------:|-------------------------:|
> > > | expert        |              -29.85  |                       -32.575 |                       -39.6   |                    -2.75 |                   -9.725 |
> > > | medium        |              -48.65  |                       -43.025 |                       -37.875 |                     5.6  |                   10.8   |
> > > | medium-expert |              -29.175 |                       -24.45  |                       -24.8   |                     4.75 |                    4.35  |
> > > | medium-replay |              -33.275 |                       -22.4   |                       -14.125 |                    10.85 |                   19.15  |
> > >
> > >
> > >
> > > > CL3: The inverted U pattern mentioned for figures 4b and 5b are very hard to grasp, and basically nonexistent for 5b unless I am parsing the sentence incorrectly.
> > >
> > > This has been edited for clarity and updated in the Appendix of the main document.

---

> > > > ### Author Response · Authors · 2021-11-23
> > > > **Response to Reviewer QjjH: continued**
> > > >
> > > >
> > > > > CL4: Figure 6 does not fully answer the question of how "important" the teacher's role is in training the student, were we expecting Transfer(0.5,0.5) to perform worse that Transfer(0, 1)? How do we maximize the teachers role in training without hurting performance?
> > > >
> > > > We found that the MMD analysis was sensitive  to the hyperparameters and couldn't get useful insights and will remove this from the main text. Instead, we studied the performance difference from teacher (in the main document and also section on the teacher policy in the overall rebuttal). Also, from the experiments on other domains, we observed that the almost always larger the weight of the teacher, the better is the performance. However, for high quality datasets, we often observe that the transfer approach does not provide much improvement, and we consider investigating this as future work.
> > > >
> > > > > More Datasets please: While this may come off as a "knee-jerk" reaction, I do believe that the class of problem defined here is important for the community. Hence it would definitely help if we are able to find real world examples of such scenarios and incorporate them here. Some new more natural additions on the dataset may include, (1) Atari benchmark dataset with semantic maps generated from internal bits. (2) D4RL dataset coupled with RGB frames, (3) Self driving / navigation dataset coupled with depth/semantic maps.
> > > >
> > > > We do agree that it is important to collect datasets on various domains in the resource constrained setting. However, due to the limited time during review we are unable to pursue this now. This is part of our future plan.
> > > >
> > > >
> > > > We sincerely hope that we were able to address the concerns about the work.

---

> > > > > ### Comment · Reviewer_QjjH · 2021-11-27
> > > > > **Thank you for the revision and addendum**
> > > > >
> > > > > I would like to to thank the authors for the clarifications and a revision of the paper that they made. However, I am leaning towards keeping my original score. The addition of predictive baseline, performance gap metrics and additional experiments provide a much better picture of their value proposition. However, the paper still lacks a strong case for the proposed method being novel or widely effective.
> > > > >
> > > > > As other reviewers have rightly pointed out, this could be a good paper with addition of well motivated datasets along with some improvement on the transfer learning algorithm so that it can perform well in datasets with good trajectories.

---

> > > > > > ### Author Response · Authors · 2021-11-28
> > > > > > **Thanks for the comment**
> > > > > >
> > > > > >
> > > > > > We thank the reviewer for the comments. We find it unfortunate that the reviewer is not willing to raise the score despite addressing most of the concerns.
> > > > > >
> > > > > > > the paper still lacks a strong case for the proposed method being novel or widely effective
> > > > > >
> > > > > > We did highlight some cases where the proposed algorithm doesn't improve over a simple baseline, which correctly shows the need for improvement of the algorithm. However, it is also important to note that we have also posed a new problem which is important for practical applications and also hard (as can be seen from the gap between the teacher and our algorithms or all the baselines considered).
> > > > > >
> > > > > > We wish the reviewer appreciates our contributions along this dimension and revises the score.

---

> > > ### Comment · Reviewer_QjjH · 2021-11-25
> > > **Thank you for the responses.**
> > >
> > > > **Response to rebuttal on W2**
> > >
> > > To clarify regarding the predictive baseline. Here, I will be assuming that $< f_{online}, f_{offline} >$ be the full set of available offline and online features. Thank you for clarifying the details regarding Transfer(1.0, 0), it was inline with my understanding that it is equivalent to TD3 + BC trained purely on the online features alone ($f_{online}$).
> > >
> > > Regarding the AutoEncoder baseline, I would like some clarity on how the auto encoder was trained and leveraged to get the prediction of $f_{offline}$ given $f_{online}$. I was assuming we could simply train a predictor network that outputs,  $f_{offline}$ given  $f_{online}$.
> > >
> > > > For the predictive baseline, we are estimating the offline state from the online state and using the predicted state for the algorithm (TD3 +BC).
> > > Yes, this is exactly how I was expecting the predictive baseline to be defined.
> > >
> > > So now the approaches for evaluation becomes [1]TD3 + BC (with only online features), [2] TD3 + BC (with both online and offline features with predictive baseline), [3] proposed approach with Transfer(0,1).
> > > I am having a hard time discerning which column refers to approach [1] and [2]. I would appreciate some clarification on this.

---

> > > > ### Author Response · Authors · 2021-11-25
> > > > **Thank you for the response. Clarification about the Table and autoencoder training**
> > > >
> > > >
> > > > We thank the reviewer for the response. Please find the table with clarified column names and added the TD3+BC (only online features) baseline.
> > > >
> > > >
> > > >
> > > > | diff          |   dim |  [1] TD3+BC (only online features)  | [2] TD3+BC (predictive features) |  [3] Transfer (0.0,1.0) |
> > > > |:--------------|------:|----------------:|-------------------:|--------------------------:|
> > > > | medium-replay |     9 |             8.7 |                8.1 |                      **11.2** |
> > > > | medium-replay |    11 |             8.3 |                8.4 |                      **11.4** |
> > > > | medium-replay |    13 |            12.1 |               12.1 |                      **15.1** |
> > > > | medium-replay |    15 |            15   |               15   |                      **18.8** |
> > > > | expert        |     9 |             7.3 |                **9.9** |                       6   |
> > > > | expert        |    11 |             9.6 |                **9.9** |                       **9.9** |
> > > > | expert        |    13 |            24   |               **24.2** |                      18.3 |
> > > > | expert        |    15 |            **32.7** |               31.5 |                      28.2 |
> > > >
> > > > ### Autoencoder training
> > > > Regarding the predictive features, we trained an autoencoder network that predicts $f_{offline}$ given $f_{online}$ which we are glad is what you wanted.
> > > > We do this by first training an autoencoder that takes the online features as input and predicts the offline features by minimizing the MSE loss between the predicted offline features and the actual offline features. The trained autoencoder is than passed to the offline RL algorithm (that is trained for deployment). During every step of training, the algorithm takes the online features, predicts the offline features using the autoencoder and uses the predicted features as the state observation. Similarly, during evaluation, the trained agent first predicts the features using online features and uses them to take an action.
> > > >
> > > > We tuned the learning rate of the autoencoder as follows. We trained an autoencoder with hidden layers [32, 64, 32] by reducing the reconstruction loss of the predicted features and the offline features on HalfCheetah-v2 reduced dimension 11 dataset. We performed a sweep for learning rate in [3e-4, 1e-4, 1e-3, 1e-2] and found 1e-3 to give the best result. We used the same hyperparameters throughout.
> > > >
> > > > Please let us know if you have any other concerns.

---

### Official Review · Reviewer_pgM4 · 2021-11-03

**Correctness:** 4
**Technical Novelty And Significance:** 3
**Empirical Novelty And Significance:** 2
**Recommendation:** 6
**Confidence:** 3

**Main Review:**

Major Comments:

As far as I know, the authors consider a novel setting where the features that exist during online deployment are more limited than in the offline dataset. This is an interesting setting and likely relevant to many applications. However, the authors choose to model this by dropping a subset of dimensions between offline and online features. I personally would appreciate some discussion on why this is practical; I would imagine that it is more realistic to assume that certain features of the state are misspecified rather than missing, as it seems to more accurately match the motivation of offline human-annotated vs online heuristically-generated features.

The authors only consider adding their transfer learning objective on top of a specific offline RL algorithm, i.e. TD3 + BC. I feel that since the novelty of the paper is in the transfer learning objective, the authors should have considered adding their objective on top of other SOTA offline RL algorithms, i.e. CQL. Right now, I feel like since the new objective was only tested on top of one specific base algorithm, the generality of the approach is more limited.

In the experiments, the authors are able to show that adding their transfer learning objective improves upon not performing any transfer learning. While this is important to show, I feel that the result is not too surprising, as you are giving a lot of additional information to the proposed algorithm. I would personally also be interested to see how the learned policy compares to the teacher policy to see how much is actually lost by using the limited features. As mentioned in the previous paragraph, I also think the paper would be improved by considering other baseline algorithms than just TD3 + BC, as it can show that the transfer learning objective can improve different classes of offline RL algorithms.

Minor Comments:

In Eq. 3 of Algorithm 1, I believe the \pi(s_i) inside the Q-function should instead be using the limited features \pi(\hat{s}_i).


**Summary Of The Paper:**

The paper proposes an offline RL algorithm in the resource-constrained setting, where the offline dataset contains richer features than provided by online interactions. The authors propose a transfer learning objective, where a teacher policy is learned from the rich features, then a policy is learned from the limited features by additionally fitting to the actions chosen by the teacher policy. The authors compare the policy learned via this transfer objective to a baseline that does not do transfer learning on D4RL tasks.

**Summary Of The Review:**

Overall, the paper makes a first step in a novel and relevant setting. The actual approach appears to be general, as the transfer learning objective can be added to any existing offline RL algorithm. And though the results in the paper are perhaps unsurprising due to how much weaker the baseline is, it is understandable because, to my knowledge, no other algorithms that could leverage the offline dataset with rich features. Hence, I recommend that it be accepted.

---

> ### Author Response · Authors · 2021-11-21
> **Response to reviewer pgM4**
>
> > I personally would appreciate some discussion on why this is practical; I would imagine that it is more realistic to assume that certain features of the state are misspecified rather than missing, as it seems to more accurately match the motivation of offline human-annotated vs online heuristically-generated features.
>
> We would like to provide an example of the case where the features are missing. The computation of some features may take more time due to sensors being power hungry such as in the case of nano-satellites (in the introduction). Additionally, auto-bidding is another example where some feature processing may be too expensive to be done online. Please refer to the auto-bidding example in the general response.
>
> We agree with the reviewer, that it is also practical to have certain features of the state misspecified rather than missing. In the experiments for ATARI 2600 (results forthcoming for the camera ready version), we provide this setup, we consider the case where the pixel input available to the agent is misspecified. More precisely, we assume that the online agent has access to a cheaper camera, therefore the image available to the online agent is a pixelated/lower resolution version of the original image available in the offline data. We hope that this addresses the feature misspecification scenario.
>
> > I feel that since the novelty of the paper is in the transfer learning objective, the authors should have considered adding their objective on top of other SOTA offline RL algorithms, i.e. CQL.
>
> For the ATARI 2600 experiments, we are considering CQL as the baseline and provide results with a transfer learning approach on top of it. Results are forthcoming.
>
> > I would personally also be interested to see how the learned policy compares to the teacher policy to see how much is actually lost by using the limited features.
>
> We will include these statistics in the main document.
>
> > In Eq. 3 of Algorithm 1, I believe the $\pi(s_i)$ inside the Q-function should instead be using the limited features $\pi(\hat{s}_i)$.
>
> Thanks for pointing this out. We will make the change in the document.
>
> We thank the reviewer for their detailed and constructive feedback. All of these points will be added to the final version of the paper as space allows.
>
> Please also review the general response at your leisure, as it contains experiments on a real-life Ads dataset.

---

> > ### Author Response · Authors · 2021-11-23
> > **Response to reviewer pgM4**
> >
> > > I feel that since the novelty of the paper is in the transfer learning objective, the authors should have considered adding their objective on top of other SOTA offline RL algorithms, i.e. CQL.
> >
> > For the ATARI 2600 experiments, we consider CQL as the baseline and provide results with a transfer learning approach on top of it. Due to the time limitations, we were unable to implement the policy regularization with CQL on continuous control experiments, and will definitely consider this for future work.
> >
> > > I feel that the result is not too surprising, as you are giving a lot of additional information to the proposed algorithm. I would personally also be interested to see how the learned policy compares to the teacher policy to see how much is actually lost by using the limited features.
> >
> > We thank you for suggesting this analysis. We do agree that the result is not too surprising. However, we have some interesting insights on what conditions allow recovering good performance by using the Transfer algorithm. We observe that when the quality of the dataset is low (medium-replay, medium), the performance recovered by using the transfer algorithm is higher, as compared to the high quality datasets (expert, medium-expert). This is a good sign, since most real world datasets are noisy (as a result of non-stationarity, or being a mixture of multiple policies) and hence  are of lower quality.
> >
> > | diff          |   baseline-teacher % |   transfer(0.5,0.5)-teacher % |   transfer(0.0,1.0)-teacher % |   diff-percent (0.5,0.5) |   diff-percent (0.0,1.0) |
> > |:--------------|---------------------:|------------------------------:|------------------------------:|-------------------------:|-------------------------:|
> > | expert        |              -29.85  |                       -32.575 |                       -39.6   |                    -2.75 |                   -9.725 |
> > | medium        |              -48.65  |                       -43.025 |                       -37.875 |                     5.6  |                   10.8   |
> > | medium-expert |              -29.175 |                       -24.45  |                       -24.8   |                     4.75 |                    4.35  |
> > | medium-replay |              -33.275 |                       -22.4   |                       -14.125 |                    10.85 |                   19.15  |
> >
> >
> > > And though the results in the paper are perhaps unsurprising due to how much weaker the baseline is, it is understandable because, to my knowledge, no other algorithms that could leverage the offline dataset with rich features. Hence, I recommend that it be accepted.
> >
> > We thank you for the detailed and constructive feedback. We hope we were able to address your concerns. We updated the document with the requested changes. Please also review the general response at your leisure, as it contains experiments on a real-life Ads dataset as well as several clarifying analyses that were requested by multiple reviewers.

---

### Official Review · Reviewer_CV2Z · 2021-11-03

**Correctness:** 4
**Technical Novelty And Significance:** 2
**Empirical Novelty And Significance:** 3
**Recommendation:** 5
**Confidence:** 4

**Main Review:**

The paper does a good job at introducing a novel scenario of resource constrained deployment, although to my mind it is a slight over-exaggeration to say that this is the “key challenge” in offline RL. The paper is very clearly written and the contributions and the experimental settings are easy to follow. The approach is simple, does not have many hyperparameters and, thus, practical. The main concern that I have is regarding the novelty of the work. While this paper talks about offline-online RL, I think very similar issues arise in the online RL or in sim2real transfer when either more features are available during the training time than at the deployment time or when in simulation we have access to the privileged information that is leveraged for the deployment. I don’t think that the resource-constraint issue is specific to the offline RL and approaches to this problem in the online and sim2real cases are not discussed. Some example of related ideas where more information is available during the training than deployment are: Cross-View Policy Learning for Street Navigation, Li et al., Privileged Information Dropout in Reinforcement Learning, Kamienny et al, Beyond Pick-and-Place: Tackling Robotic Stacking of Diverse Shapes. Lee at al., Continual Reinforcement Learning deployed in Real-life using Policy Distillation and Sim2Real Transfer, Traoré et al. Could the authors provide a more detailed overview of the related work in the online and sim2real RL? Are there other baselines that could be relevant to compare against? Then, it seems to me that the main contribution of the paper is rather empirical. The experiments are well designed, explained and executed on three control environments. However, the diversity of the environments is rather limited and I would like to see a broader set of experiments on varying tasks to appreciate the benefits of the method (maybe training policies from vision? maybe some discrete control? maybe some navigation tasks?)

Pros:

- Interesting and well motivated setting of resource constrained deployment.

- Simple and clear approach.

- I appreciated the dataset construction component that helps to test the proposed algorithm in more realistic settings.

Cons:

- Related literature is not covered completely. How similar problems are addressed in online RL and sim2real?

- Limited novelty of the proposed approach.

- The experiments are only conducted on a small set of environments of limited diversity.

Other points and questions:

- The way I read figure 1b only demonstrates that more features is better. It does not necessarily show that the limitation comes because of ignoring privileged information, it may well be just because limited features are available at the deployment time.

- Do the authors mean “policy improvement step” instead of “policy iteration step” in the algorithm description?

- Percentage of seeds better than the baseline is an unusual metric, what is the motivation of it? What is the advantage against a more standard percentage of improvement?

- Could the authors explain better why the gain with a small number of available features is smaller or even negative? I understand that the performance in general would degrade, but I would expect that the baseline would suffer even more.


**Summary Of The Paper:**

This paper presents a new under-explored problem in offline RL: the situation when during the online deployment some features that were present in the offline dataset are missing. The authors motivate this problem by the challenges in real applications. They demonstrate that a straightforward approach of training an offline policy on a set of restricted features suffers from a loss in performance. Then, they propose an extension of an existing offline RL algorithm to distil the teacher offline policy which has access to all features to a student policy which has access to a limited feature set. Finally, the authors conduct a set of experiments on 3 Mujoco control tasks where they vary different parameters of the problem, for example, the quality of the datasets (and the way they were collected), the number of dropped dimensions in the resource-constrained setting. The results show the benefits of the proposed method compared to the baseline.

**Summary Of The Review:**

This paper proposes a simple but efficient approach to the resource constrained deployment of offline RL policies. I would like to hear more about the contributions of this work compared to prior works in similar settings in online RL and sim2real. I appreciate the experiments and new dataset construction, but I would like to see a broader set of experiments on a larger variety of tasks to be able to make conclusions about the efficiency of the method.

---

> ### Author Response · Authors · 2021-11-21
> **Response to reviewer CV2Z**
>
> ## Literature review of similar work
>
> We thank the reviewer for bringing to our attention various works in the online setting which also distill knowledge from an unconstrained teacher into a constrained student. We have carefully reviewed the literature in this area and have made the following observations:
>
> * Cross-View Policy Learning for Street Navigation, Li et al.: The paper uses two different views (aerial view and ground view) when training a navigation model, but sometimes only the aerial view is available during testing.
> * Privileged Information Dropout in Reinforcement Learning, Kamienny et al: The paper studies a path finding task to reach to a goal. The agent only has access to egocentric observation (i.e., it only sees nearby area, not the entire map). The privileged information contains the entire map with the goal.
> * Beyond Pick-and-Place: Tackling Robotic Stacking of Diverse Shapes, Lee at al. and  Continual Reinforcement Learning deployed in Real-life using Policy Distillation and Sim2Real Transfer, Traoré et al. These papers study the sim2real setting. They aim to train models that control a robot, and the simulator can very accurately simulate the robot movement. Both papers have access to a very good simulator. Unlike these methods, we don't assume that we have access to an accurate simulator and we only have limited offline data available which is collected using some known or unknown policy.
>
> All of these works either require access to a high-quality simulator at some point during training or have sufficient resources to train models online. In real applications either of these may not be available, but in general offline data from arbitrary policies is available. One challenge that arises from this is that the teacher used for transfer (in the above works) generally performs well, whereas in our setting, the teacher itself may be imperfect depending on the quality of the data. We focus on this setting, and believe it is an important and distinct addition to the literature. However, these papers should all be critical elements of our literature review and we will include them in the final draft.
>
>
>
>
>
> > The way I read figure 1b only demonstrates that more features is better. It does not necessarily show that the limitation comes because of ignoring privileged information, it may well be just because limited features are available at the deployment time.
>
> We wanted to simulate the setup of privileged information (or lack thereof) by removing some features from the available observation space. In Figure 1b, therefore, we consider the missing features as privileged information which otherwise is unavailable in the online setting. The message might have come off as 'more features is better', but that is due to the setup where we simulated the lack of privileged information with missing features.
>
> > Do the authors mean “policy improvement step” instead of “policy iteration step” in the algorithm description?
>
> Yes, policy improvement is more appropriate, and we will change this.
>
> > Percentage of seeds better than the baseline is an unusual metric, what is the motivation of it? What is the advantage against a more standard percentage of improvement?
>
> We provide results for both percentage of seeds better than baseline, and percentage improvement. For a given setting (dataset) and a given seed, everything else between the baseline and the proposed algorithm is the same (since all hyperparameters are same). So we wanted to emphasize the success rate of using the proposed method. Percentage improvement only provides a point estimate which may be skewed due to few extreme data points. For example, consider the results for RC-D4RL on `Walker2d-v2, medium-replay, dim:9`. The performance of the `basline` is already small $(0.8\pm 1.5)$, and `Transfer (0.0,1.0)` achieves $4.1\pm 5.4$, which is around 400% improvement over the baseline, but the absolute performance is still bad. For the sake of completeness, we included both results to give a full picture of the improvement.
>
> > Could the authors explain better why the gain with a small number of available features is smaller or even negative? I understand that the performance in general would degrade, but I would expect that the baseline would suffer even more.
>
> We believe this is because for the transfer algorithm to work effectively, there needs to be a correlation between the observations that are presented to the teacher and to the student. The more the correlation, the larger the performance gain from transfer. With fewer features in the online case, the observations of the student and the teacher are less correlated and therefore, the action suggested by the teacher policy may not be effectively distilled to the student. In such cases the transfer may sometimes be harmful.
>
> We thank the reviewer for their detailed and constructive feedback, and will include all these points in the final version of the paper as space allows.

---

> > ### Author Response · Authors · 2021-11-21
> > **addendum**
> >
> > Please also review the general response at your leisure, as it contains experiments on a real-life Ads dataset.

---

> > ### Comment · Reviewer_CV2Z · 2021-11-22
> > **Thank you for the responses**
> >
> > I would like to thank the authors for their responses.
> >
> > Regarding the literature review.
> >
> > Thank you for the description of the papers that I mentioned. Of course, I realize that the settings are quite different between the works that were mentioned and your work. However, my point was that beyond the difference in the setting, the *techniques that are used to adapt the policy* in a situation with less features could be quite similar. I would recommend focusing the literature review on this aspect. Besides, this was not in any way an exhaustive list of works and you may find other related methods as well.
> >
> > Regarding the request for the more diverse environments.
> >
> > I can see that all other reviewers raised similar concerns, thus, I believe it is an important point. Thank you for providing a new baseline with Adds. I think in the text, it would be nice to describe it as MPD: what is the state, action, reward etc. Moreover, I currently do not understand why this environment is a full MPD and not just a bandit with arm features.

---

> > > ### Author Response · Authors · 2021-11-23
> > > **second response to reviewer CV2Z**
> > >
> > > > Literature review
> > >
> > > Thanks for clarifying your point on the literature review. We do agree that different knowledge distillation techniques can be applied to our setting. For example, Li et al. (2019) train a model so that features from different domains have similar embeddings, and Kamienny et al. (2020) perturb the feature using a random noise centered at the privileged information. These are all interesting future directions of resource-constrained offline RL. However, the papers related to Sim2Real (Lee et al., 2021; Traor ́e et al., 2019) require access to an accurate simulator, and the interaction with the simulator seems to be crucial. We will make these points clear in the new version of the draft. Lastly, we do acknowledge that these papers by no means cover all the areas, and thus already added a few good survey papers (e.g., Zhu et al., 2020; Wang & Yoon, 2021). If you have better references, we'd be happy to include them in the paper.
> > >
> > > > I can see that all other reviewers raised similar concerns, thus, I believe it is an important point. Thank you for providing a new baseline with Adds. I think in the text, it would be nice to describe it as MPD: what is the state, action, reward etc. Moreover, I currently do not understand why this environment is a full MPD and not just a bandit with arm features.
> > >
> > > Along with Ads results, we added new results on Atari-2600 datasets (discrete control task).
> > > As for the MDP for Ads, please review both the final paper draft section 5.3 and the overall rebuttal (Evaluation on Ads data: Updated) for a detailed explanation on why it is not a contextual bandit.
> > >
> > > Here is the summary of why MDP formulation: "At each time step, an auto-bidding agent has some information available like time of the query, details of the ad text or bidded keyword, available budget, past spend, model-based estimate of how likely the consumer is to click an ad, etc. This information is used as a state. Based on the state, the agent takes a action (decides the bid) and gets feedback (if the ad was selected and/or clicked, and how much the click costs). If the ad was clicked, the "available budget" feature in the state is updated. The budget constraint in this problem means that it cannot be modeled using contextual bandits (as many Ads problems can), because the next state depends on the action taken."
> > >
> > > We hope that we were able to address your concerns about the paper.

---

> > > > ### Comment · Reviewer_CV2Z · 2021-11-26
> > > > **Thanks for clarification and revision**
> > > >
> > > > I would like to to thank the authors for the clarifications and a revision of the paper that they made. However, I am still willing to keep my original score.
> > > >
> > > > I liked the addition of the new baseline with the predictive model and the discrete control in atari. While additional experiments are a great addition, the results are somehow mixed as mentioned by reviewer oGRd. It would also be nice to see some baselines inspired by the methods from sim2real and online RL, this would support the claims of the novelty in this work.

---

> > > > > ### Author Response · Authors · 2021-11-28
> > > > > **Thanks for the comments**
> > > > >
> > > > >
> > > > > We thank the reviewer for the comments. However, we find it unfortunate that the reviewer is not willing to raise the score despite addressing most of the concerns.
> > > > >
> > > > > > the results are somehow mixed...
> > > > >
> > > > > We do agree that there are cases where a simple baseline overcomes the proposed method, however the gap between the teacher and these baselines is also very high. This shows that the new problem setting that we proposed in this paper is hard in addition to being important for practical applications, and that this is an important and challenging problem for the community.
> > > > >
> > > > > We sincerely hope that the reviewer appreciates our contributions along this dimension and revises the score.

---

### Author Response · Authors · 2021-11-21
**Overall comments to all reviewers**

# Overall

## Additional simulated environments
We thank the reviewers for suggestions on additional RL settings on which to evaluate our algorithm. We are currently running Atari experiments to further test the robustness of our results. Results are forthcoming by the end of the response period. As these experments finish, we will share the results here and also add them into the main paper (as space allows).


## Evaluation on Ads data
Thanks to the reviewers for their constructive feedback. We can fully appreciate that purely simulated settings may not be representative for evaluating an algorithm which is meant for real-world application. Thus, we have studied agent performance in the auto-bidding task for online advertising. Online advertising is a dynamic, stochastic environment. Advertisers have increasingly delegated decisions to machines in order to achieve increased return on investment. Auto-bidding is one of the critical components in this shift towards AI-driven optimization. Auto-bidding agents determine a unique bid for each opportunity in real-time. This problem is one of distributed, stochastic control in a partially observable, stochastic, non-stationary environment [1]. This type of environment is extremely difficult to develop complex algorithms for, and it demonstrates our setting well in that there are a myriad of computational constraints that limit the type of models that can be considered during online operation.

In this task, agents attempt to maximize the number of ad clicks they collect during a day by bidding on queries (on which ads are shown). The set of queries which receive a bid are set by the advertiser's choice of bidded keywords. Agents are given a fixed daily budget to do this. They are charged according to some blackbox auction mechanism only if their ad is clicked. Thus, agents must balance between saving budget for future opportunities, and buying guaranteed ad space now.

We have pulled query-level Ads data for 10 days and 8,000 advertisers (from a proprietary dataset). Features in this data set include
1. The time of the query
2. A model-based estimate of how likely the consumer is to click an ad. Includes ad and user data.
3. Model-based embeddings of the query.
4. Model-based embeddings of the bidded keyword.

1 and 2 are examples of features that are available online and offline, as they are easy and fast to observe. 3 and 4 are examples of features that are only available offline, as they require extensive query times which are infeasible in this setting. There are a total of about 881 features available offline and 111 features available online. The main constraint limiting the types of models and features used in the online setting is computation time, as the user who generated the query will not accept a wait time of more than a fraction of a second before they want to see their search results.

We trained on 80% of the advertisers using the proposed algorithm in the paper and evaluated on the remaining 20%. For evaluation, we implemented a Q-value approximation using FQE [2]. The results are as follows:

# [Figure Link](https://i.imgur.com/xE1AgiS.png)


[1] Bottou, Léon, et al. "Counterfactual Reasoning and Learning Systems: The Example of Computational Advertising." Journal of Machine Learning Research 14.11 (2013).

[2] Le, Hoang, Cameron Voloshin, and Yisong Yue. "Batch policy learning under constraints." In International Conference on Machine Learning, pp. 3703-3712. PMLR, 2019.


In the figure (please click the link), FQE estimates are normalized by the value estimate at the beginning of the first epoch (before policies are trained), which is the same value for all settings (since before training all policies are initialized as random with a fixed seed). Normalization is done to mask the identities of advertisers, which are business-sensitive. The model with a higher $\beta_2$ value tends to perform better than the ones with lower values. Recall that $\beta_2$ is the weight for the transfer learning, (higher $\beta_2$ means more transfer is incorporated in the objective), this trend suggests that there is a benefit to incorporating transfer learning. This plot used a single seed value, but in the camera-ready version, we will run this experiment using multiple seeds and then provide error bars. Also, we will run a statistical test on the rankings of the five algorithm settings in the above figure.

We welcome discussion on these results and will include them in the final version of the paper with explanations.

---

### Author Response · Authors · 2021-11-23
**Experiments on ATARI 2600**

We thank the reviewers for suggestions on additional RL settings on which to evaluate our algorithm. Below are additional results which we believe address these concerns.

## Setup
We evaluate our proposed approach on the Atari 2600 suite of games, which is a discrete control problem that is challenging especially due to the high dimensional visual input and delayed credit assignment. Since we are in the offline RL setting, we use the DQN replay dataset from Agarwal et al [1] which is a standard benchmark to evaluate offline RL algorithms for discrete control problems. In particular, we use the data for the following games: Pong, Qbert, Breakout.

### Datasets
The DQN replay dataset consists of 50M state transitions that are collected during the training of a DQN agent in the online setting. Each transition consists of a tuple (state, action, reward, next_state) where the state and next state are 84x84 images and the action space is discrete. Following earlier works, we consider environments with sticky actions (the agent takes current action with probability 0.25, or otherwise repeats past actions). Additionally, we use stacking of 4 consecutive frames together as the state space which is a common technique used for the Atari suite. Instead of using the full 50M transitions, we use 1M transitions in the dataset that are collected during the end of the DQN online training. These transitions contain expert level trajectories since they are collected during the end of training. We use a Discrete version of CQL [2] (which is a state of the art algorithm for Atari offline RL) as a baseline, and implement a transfer algorithm on top of it.

To simulate the online and offline features, we consider the following setup. We assume the actual image observation (4x84x84) as the offline features, and a pixelated version of the images as online features (this setting can simulate sensors of different resolution). To simulate this, we resize each (84x84) image to 16x16 and then resize it back to 84x84. The description of the transfer algorithm is as follows.

### Algorithm

Please see Section 4.2 in the paper.

## Training
We trained the algorithms with a batch size of 32 for 1M batches  using the same network architecture as the DQN (N-DQN) [3] (which is also used in Agarwal[1], CQL[2],etc). Nature DQN uses the following network architecture
```
[32, 8, 4], [64,4,2], [64,3,1], 512
```
 where `512` is the dimension of the final layer. We consider the CQL agent using N-DQN as the Baseline, and we denote the proposed algorithm as Transfer agent. Similar to CQL, we used the quantile regression DR-DQN to compute the DQN loss.

In order to simulate the power constrained setting (as in the nano-satellite example), we also consider the setup where the agent to be deployed online uses a different network architecture than the teacher network architecture (N-DQN). This enables the agent to utilize lesser memory and power during deployment. We call this network architecture as (S-DQN) by changing the size of features in the final layer. Specifically, we vary the feature dimension with values `256, 128, 64`. We can this setting the power constrained setting. Below, we show the number of parameters of each network.

| Network (feature size) | #Parameters |
|:---------|---------------:|
| N-DQN (512) | 1.6 M |
| S-DQN (256) | 0.88 M |
| S-DQN (128) | 0.48 M |
| S-DQN (64) | 0.28 M |

We evaluate during training at a frequency of every 20k batches by assuming access to the environment (assumption that is made in [1,2]), and consider the score as the average reward achieved in the last 10 evaluations. We evaluate the teacher on the environment using full offline features, whereas we evaluate the Baseline and the Transfer agents using the pixelated images.  We trained the algorithms for 3 seeds and provide the mean and std of the results. We also show the human/gamer score as a reference from [4].

We selected the value of `beta` by choosing the value that resulted in the best performance on `Qbert` and used it for the other environments. We did this separately for the N-DQN setting and the power constrained setting.


We observe from both the settings that there is a significant drop in performance of the Baseline and Transfer agents using the online features as compared to the Teacher. The Transfer agent performs marginally better than the Baseline agent on all the three games for the N-DQN setting. For the S-DQN setting, the Transfer agent outperforms the Baseline for Pong on all the three encoder sizes considered. For Qbert, the Transfer agent outperforms the Baseline for one of the encoder sizes.

It is important to highlight that there is a significant gap between the performance of the Teacher and the Transfer/Baseline agents for both the settings. This suggests that more tailored transfer learning approaches are required for the Resource Constrained Offline RL problem.

---

> ### Author Response · Authors · 2021-11-23
> **Experiments on ATARI 2600**
>
> ## Results
> ### Using N-DQN
>
> | Env      |   Teacher mean (N-DQN) |   Teacher std (N-DQN) |   Transfer mean (N-DQN) |   Transfer std (N-DQN) |   Baseline mean (N-DQN) |   Baseline std (N-DQN) | Gamer |
> |:---------|---------------:|--------------:|----------------:|---------------:|----------------:|---------------:|---:|
> | pong     |           8.2  |          1.89 |           -1.02 |           1.1  |           -2.81 |           1.39 | 15 |
> | qbert    |        7890.25 |       1896.22 |         6343.92 |         346.45 |         5833.58 |         584.74 | 13455|
> | breakout |         106.63 |          8.64 |            6.07 |           0.08 |            4.65 |           0.22 | 30|
>
>
> ### Power Constrained (using S-DQN)
> |Env   |   Encoder |   Transfer mean (S-DQN) |   Transfer std (S-DQN) |   Baseline mean (S-DQN) |   Baseline std (S-DQN) |
> |:------|----------:|----------------:|---------------:|----------------:|---------------:|
> | pong  |        64 |           -3.53 |           5.01 |           -5.25 |           7.95 |
> | pong  |       128 |           -1.83 |           4.75 |           -7.99 |           1.52 |
> | pong  |       256 |           -1.08 |           0.88 |           -1.74 |           4.83 |
> | qbert |        64 |         1336.85 |         566.79 |         1830    |         908.12 |
> | qbert |       128 |         2524.8  |         717.82 |         2808.92 |        1577.74 |
> | qbert |       256 |         4762.95 |         542.23 |         4710    |          73.86 |
>
>
> [1] Agarwal, Rishabh, Dale Schuurmans, and Mohammad Norouzi. "An optimistic perspective on offline reinforcement learning." International Conference on Machine Learning. PMLR, 2020.
>
> [2] Kumar, Aviral, et al. "Conservative q-learning for offline reinforcement learning." arXiv preprint arXiv:2006.04779 (2020).
>
> [3] Mnih, Volodymyr, et al. "Human-level control through deep reinforcement learning." nature 518.7540 (2015): 529-533.
>
> [4] Chen, Lili, et al. "Decision transformer: Reinforcement learning via sequence modeling." arXiv preprint arXiv:2106.01345 (2021).

---

### Author Response · Authors · 2021-11-23
**Additional baselines (MuJoCo Experiments)**

We thank the reviewers for suggesting a wider range of baselines. We believe that the following experiments, included in the paper, will help to address these suggestions.

## Pure Behavior Cloning Transfer (From Teacher)

See Section C.4.1 in the paper.

## Predictive Model:

We also consider an additional baseline that predicts the missing features (offline features) from the available online features. We do this by first training an autoencoder that takes the online features as input and predicts the offline features by minimizing the MSE loss between the predicted offline features and the actual offline features. The trained autoencoder is than passed to the offline RL algorithm (that is trained for deployment). During every step of training, the algorithm takes the online features, predicts the offline features using the autoencoder and uses the predicted features as the state observation. Similarly, during evaluation, the trained agent first predicts the features using online features and uses them to take an action.

We tuned the learning rate of the autoencoder as follows. We trained an autoencoder with hidden layers `[32, 64, 32]` by reducing the reconstruction loss of the predicted features and the offline features on `HalfCheetah-v2` reduced dimension `11` dataset. We performed a sweep for learning rate in `[3e-4, 1e-4, 1e-3, 1e-2]` and found `1e-3` to give the best result. We used the same hyperparameters throughout.


We evaluate the algorithm on HalfCheetah environment in the RC-D4RL datasets, and summarize the results in the following table. The training and evaluation procedure is similar as the main experiments.

## Results

| Difficulty          |   Dimension |   True-bc |   Autoencoder |   Transfer (0.0,1.0) |
|:--------------|------:|---------------:|-------------------:|--------------------------:|
| medium-replay |     9 |            8.1 |                8.1 |                      **11.2** |
| medium-replay |    11 |            8.1 |                8.4 |                      **11.4** |
| medium-replay |    13 |           13.9 |               12.1 |                      **15.1** |
| medium-replay |    15 |           13.9 |               15   |                      **18.8** |
| expert        |     9 |           **13.1** |                9.9 |                       6   |
| expert        |    11 |           **14.4** |                9.9 |                       9.9 |
| expert        |    13 |           **24.8** |               24.2 |                      18.3 |
| expert        |    15 |           **34**   |               31.5 |                      28.2 |



### Summary

| Difficulty          |  Avg % Diff over True BC | Avg % Diff over Autoencoder |
|:--------------|------------------:|--------------------:|
| expert        |           -32.175 |             -18.575 |
| medium-replay |            30.725 |              31.025 |

We can observe from the results that the True BC agent is very effective in expert dataset (which is of high quality) whereas the proposed algorithm is effective in medium-replay dataset (which is of low quality). It is most often the case that real world datasets are of poor quality due to several factors such as non-stationarity, interference, etc. Similar conclusions hold for the predictive baseline, but using an autoencoder for the predictive baseline may also increase computation on the agent during deployment. From these results, we believe that future work is needed to develop stronger transfer learning algorithms tailored to the RC Offline RL framework.

---

### Author Response · Authors · 2021-11-23
**Teacher Analysis (MuJoCo Experiments)**


We thank the reviewers for suggesting an analysis on the performance of the teacher policy against the learned policy.

Here, we present an analysis of the drop in performance of using only online features (in the baseline) by measuring the percentage difference as compared to the teacher model (trained using full offline features). We consider the RC-D4RL HalfCheetah-v2 for this analysis. The table shows that the baseline consistently underperforms the teacher (as far as -70% in some cases). We also show the percentage drop in performance of the proposed transfer method trained using the teacher. We can still see some cases where the transfer model significantly underperforms the teacher. However, by using the teacher model during transfer, the proposed model recovers a lot of performance (as compared to the baseline using only online features).

We would like to highlight that despite the transfer algorithm recovering some performance as compared to baseline, it still suffers noticeable drops in performance as compared to the teacher. We believe that more tailored approaches to the Resource Constrained Offline RL framework need to be developed to bridge this gap.


| diff          |   dim |   baseline-teacher % |   transfer (0.0, 1.0) -teacher % |   recovered performance |
|:--------------|------:|---------------------:|---------------------:|---------------:|
| medium-replay |     9 |                -36   |                -17.6 |           18.4 |
| medium-replay |    11 |                -40.3 |                -18   |           22.3 |
| medium-replay |    13 |                -35.3 |                -19.3 |           16   |
| medium-replay |    15 |                -21.5 |                 -1.6 |           19.9 |
| medium        |     9 |                -77.9 |                -56.6 |           21.4 |
| medium        |    11 |                -67.1 |                -60.5 |            6.6 |
| medium        |    13 |                -37.2 |                -30.1 |            7.1 |
| medium        |    15 |                -12.4 |                 -4.3 |            8.1 |
| medium-expert |     9 |                -43.2 |                -37.5 |            5.7 |
| medium-expert |    11 |                -43.6 |                -33.3 |           10.3 |
| medium-expert |    13 |                -26.3 |                -16.1 |           10.2 |
| medium-expert |    15 |                 -3.6 |                -12.3 |           -8.8 |
| expert        |     9 |                -56.5 |                -64.3 |           -7.7 |
| expert        |    11 |                -41.1 |                -39.3 |            1.8 |
| expert        |    13 |                -15.5 |                -35.6 |          -20.1 |
| expert        |    15 |                 -6.3 |                -19.2 |          -12.9 |

---

### Author Response · Authors · 2021-11-23
**Evaluation on Ads data: Updated**

Thanks to the reviewers for their constructive feedback. We can fully appreciate that purely simulated settings may not be representative for evaluating an algorithm which is meant for real-world application. Thus, we have studied agent performance in the auto-bidding task for online advertising. Online advertising is a dynamic, stochastic environment. Advertisers have increasingly delegated decisions to machines in order to achieve increased return on investment. Auto-bidding is one of the critical components in this shift towards AI-driven optimization. Auto-bidding agents determine a unique bid for each opportunity in real-time. This problem is one of distributed, stochastic control in a partially observable, stochastic, non-stationary environment [1]. This type of environment is extremely difficult to develop complex algorithms for, and it demonstrates our setting well in that there are a myriad of computational constraints that limit the type of models that can be considered during online operation.

In this task, agents attempt to maximize the number of ad clicks they collect during a day by bidding on queries (on which ads are shown). The set of queries which receive a bid are set by the advertiser's choice of bidded keywords. Agents are given a fixed daily budget to do this. They are charged according to some black-box auction mechanism only if their ad is clicked. Agents must balance between saving budget for future opportunities, and buying guaranteed ad space now. At each time step, an auto-bidding agent has some information available like time of the query, details of the ad text or bidded keyword, available budget, past spend, model-based estimate of how likely the consumer is to click an ad, etc. This information is used as a state. Based on the state, the agent takes a action (decides the bid) and gets feedback (if the ad was selected and/or clicked, and how much the click costs). If the ad was clicked, the "available budget" feature in the state is updated. The budget constraint in this problem means that it cannot be modeled using contextual bandits (as many Ads problems can), because the next state depends on the action taken. If the agent bids USD 10 given an available budget of USD 100, it can expect to have at least USD 90 to spend for the rest of the day. This USD 90 is part of the state for the next step. If the agent bids USD 1 in the same situation, it may expect to have at least USD 99 for the rest of the day. This rest-of-day budget constraint is a key part of the state, as it restricts the trajectory of future states and actions. Thus, we need an MDP formulation to tackle this problem.


We have pulled query-level Ads data for 10 days and 8,000 advertisers (from a proprietary dataset). Features in this data set include
1. The time of the query
2. A model-based estimate of how likely the consumer is to click an ad. Includes ad and user data.
3. Available budget
4. Past spend
5. Model-based embeddings of the query.
6. Model-based embeddings of the bidded keyword.

1,2,3 and 4 are examples of features that are available online and offline, as they are easy and fast to observe. 5 and 6 are examples of features that are only available offline, as they require extensive query times which are infeasible in this setting. There are a total of about 881 features available offline and 111 features available online. The main constraint limiting the types of models and features used in the online setting is computation time, as the user who generated the query will not accept a wait time of more than a fraction of a second before they want to see their search results.

We trained on 80% of the advertisers using the proposed algorithm in the paper and evaluated on the remaining 20%. For evaluation, we implemented a Q-value approximation using FQE [2]. The results are as follows:

# [Figure Link](https://i.imgur.com/VD2I93E.png)

---

> ### Author Response · Authors · 2021-11-23
> **Evaluation on Ads data**
>
>
> In the figure, FQE estimates are normalized by the value estimate at the beginning of the first epoch (before policies are trained), which is the same value for all settings (since before training all policies are initialized as random with a fixed seed). Ten different seeds are used, and then the normalized metrics are aggregated by average. Normalization is done to mask the identities of advertisers, which are business-sensitive. Transfer (0.0, 1.0) tends to perform better than other models including the baseline, Transfer (1.0, 0.0), without transfer. Furthermore, out of 10 different seeds, Transfer (0.0, 1.0) outperforms Transfer (1.0, 0.0) 7 times. This trend suggests that there is a benefit to incorporating transfer learning.
>
> We welcome discussion on these results and they are included in the main document (Section 5.3).
>
> [1] Bottou, Léon, et al. "Counterfactual Reasoning and Learning Systems: The Example of Computational Advertising." Journal of Machine Learning Research 14.11 (2013).
>
> [2] Le, Hoang, Cameron Voloshin, and Yisong Yue. "Batch policy learning under constraints." In International Conference on Machine Learning, pp. 3703-3712. PMLR, 2019.

---

### Author Response · Authors · 2021-11-23
**Summary Response**

We thank the reviewers for suggestions on the experiments and additional RL settings to evaluate our algorithm. We have incorporated most of the suggestions possible during this timeframe. The following are the main additions to the document, these are explained in individual responses below.

- Evaluation on Ads data: We showed results on a real life Ads (autobidding) dataset in Section 5.3 of the main document.
- Teacher Analysis (MuJoCo Experiments): We performed additional analysis on the MuJoCo environments to better understand the resource constrained setting. In particular, we studied the performance loss (compared to Teacher) by not using offline features (for the baseline) and the amount of performance recovered by the transfer learning approach (See Section 5.1.2, Table 2, 5).
- Additional baselines (MuJoCo Experiments): We studied two additional baselines suggested by the reviewers (True-BC and Predictive baseline) and compared with our algorithm and provided insights.
- Experiments on ATARI 2600: To bring diversity into the problems considered, we show the results on Atari 2600 environments (discrete control task) in Section 5.2 of the main document.

We hope the updated results alleviate the reviewers' concerns about the work, and welcome discussion on these changes.

---

### Decision · Program_Chairs · 2022-01-20

**Decision:**

Reject

**Comment:**

The authors propose the resource constrained offline RL problem where the offline dataset contains extra features that are not available online. The goal is to use these extra features to improve performance during deployment. They propose a simple modification to TD3-BC in the continuous control setting and a simple modification to CQL in the discrete setting. They evaluate their proposed approaches on D4RL, RC-D4RL (a novel dataset that they introduce for resource constrained offline RL), Atari, and a proprietary real-life Ads problem.

Initial reviews identified the following concerns:
* While the exact problem is novel, the idea of having access to privileged features at training time that are not available at deployment has been explored in supervised learning and online RL. The reviewers were not clear how considering the offline RL setting interacts specifically with the privileged features to produce an interesting setting.
* The baseline simply trains on the limited feature set. Unsurprisingly, using the extra features can improve performance. In light of the previous point, reviewers asked for more substantial baselines, suggesting BC on the teacher and predicting the missing features as some possibilities.
* The set of tasks was too limited.

The authors provided a substantial response:
* Experiments on Ads data
* Experiments on Atari with CQL as the base algorithm
* Additional baselines on RC-D4RL HalfCheetah-v2 datasets (BC on teacher and predictive)
* Additional analysis

I commend the authors on the hard work they did preparing this response. It is quite substantial and does improve the paper significantly. However, reviewers and I still have a number of concerns:
* The additional baselines are appreciated, however, the results are mixed. The additional baselines are a step in the right direction, but they need to be evaluated beyond a single dataset. It is hard to evaluate the results without reasonable baselines. I agree and think that even though the specific problem is novel, the idea of transfer learning is not, so it is reasonable to require that we have more extensive baselines. Furthermore, while the authors argue that their method has an edge on the more practical dataset, that is based on a very limited evaluation. Probing this further is important.
* The CQL modification is quite different than the TD3+BC modification. The performance of the modification for CQL is not significantly better than CQL. What should we make of this?
* For the Ads dataset, all hyperparameter settings except Transfer(0, 1) show the same performance. This seems surprising as even Transfer(0.1, 0.9) shows no difference. Finally, Transfer(0, 1) beating Transfer(1, 0) 7/10 times is not statistically significant.

At this time, the paper is not ready for publication, but the paper is moving in the right direction and I encourage the authors to submit a revised version to a future venue.